# Approximate Nearest Neighbor Search for Modern AI: A Projection-Augmented Graph Approach

**Kejing Lu** [1]  **Zhenpeng Pan** [2]  **Yoshiharu Ishikawa** [3]  **Chuan Xiao** [4][3]  **Jianbin Qin** [5]

## Abstract

Approximate Nearest Neighbor Search (ANNS) is fundamental to modern AI applications. Most existing solutions optimize query efficiency but fail to align with the practical requirements of modern workloads. In this paper, we outline six critical demands of modern AI applications: high query efficiency, fast indexing, low memory footprint, scalability to high dimensionality, robustness across varying retrieval sizes, and support for online insertions. To satisfy all these demands, we introduce Projection-Augmented Graph (PAG), a new ANNS framework that integrates projection techniques into a graph index. PAG reduces unnecessary exact distance computations through asymmetric comparisons between exact and approximate distances as guided by projection-based statistical tests. Three key components are designed and integrated into the graph index to optimize indexing and searching. Experiments on six modern datasets demonstrate that PAG consistently achieves superior queries per second (QPS)-recall performance—up to $5\times$ faster than HNSW—while offering fast indexing speed and moderate memory footprint. PAG remains robust as dimensionality and retrieval size increase and naturally supports online insertions. Our source code is available at: https://github.com/KejingLu-810/PAG/.

## 1. Introduction

Given a dataset $\mathcal{D} \subset \mathbb{R}^d$ with size $n$ and a query $q \in \mathbb{R}^d$, Approximate Nearest Neighbor Search (ANNS) aims to find $K$ approximate nearest neighbors of $q$ as accurately and efficiently as possible. Due to its crucial role in many applications such as image search, recommender systems, and RAG, we have witnessed the rapid proliferation of ANNS solvers. Whereas the ANN-Benchmarks (Bernhardsson et al., 2015) has been developed to compare these methods in a unified environment, many datasets in the ANN-Benchmarks, as pointed out by Chen et al. (2025), are derived from outdated models such as SIFT and GIST, which are image descriptors developed more than 20 years ago (Lowe, 2004; Oliva & Torralba, 2001). In addition, its evaluation metric is limited to QPS-recall under $K = 10$.

Seeing the fast development of AI technology, we argue that a well-designed ANNS solver should perform well on modern datasets, and beyond **(D1) QPS-recall** performance, several demands must also be considered: **(D2) indexing time**, for which graph indexes such as HNSW (Malkov & Yashunin, 2020) are often criticized (Lin, 2025; Li & Papakonstantinou, 2025), compromising their use in applications that require instant deployment; **(D3) memory footprint**, where moderate and adjustable consumption, mostly caused by the index size, is desired for the memory vs accuracy trade-off (Cheng et al., 2024; Wallace, 2025); **(D4)** scalability to **high dimensionality**, as motivated by the increasing dimensionality of modern embedding models such as CLIP (Radford et al., 2021) [1]; **(D5)** robustness against the **retrieval size** $K$, due to the needs in various applications (e.g., $K$ is typically 10 for RAG (Fensore et al., 2025; Ke et al., 2025) but can be up to hundreds in image retrieval (Vendrow et al., 2024) and thousands in recommender systems (Zhang et al., 2025)); **(D6)** support of **online insertions**, which is essential for emergent applications such as self-evolving agents that continually accumulate and reuse experience through interaction (Ouyang et al., 2025; Zhai et al., 2025; Zhang et al., 2026).

### 1.1. Prior Works

We review representative ANNS solvers based on D1 – D6. More discussions on literature can be found in Appendix C.

---

[1]University of Yamanashi [2]Beijing Institute of Technology, Zhuhai [3]Nagoya University [4]Osaka University [5]Shenzhen University. Correspondence to: Jianbin Qin <jqin@szu.edu.cn>.

*Proceedings of the 43rd International Conference on Machine Learning*, Seoul, South Korea. PMLR 306, 2026. Copyright 2026 by the author(s).

---

[1]Despite the availability of dimensionality reduction techniques, Weller et al. (2025) proved that for embedding-based retrieval with dimensionality $d$, there exists a dataset size $n$ (e.g., $n \approx$ 4M for $d = 1024$) beyond which it is theoretically impossible to represent all possible top-2 document combinations, showcasing the necessity for higher dimensionality on large datasets.

*Table 1.* Summary of empirical performance comparison of notable ANNS solvers, HNSW (Malkov & Yashunin, 2020), Vamana (in-memory DiskANN (Subramanya et al., 2019)), IVFPQFS (fast scan IVFPQ (Jégou et al., 2011)), ScaNN (Guo et al., 2020a), RaBitQ+ (Gao et al., 2025), SymQG (Gou et al., 2025a), HNSW+KS2 (Lu et al., 2025), and our solutions PAG-Base (for high QPS) and PAG-Lite (for fast indexing and small index size). Detailed results are available in Sec. 5.2. The evaluation of D1 – D3 is clustered into tiers, where Tier-4 is the best. D4 – D6 are evaluated in yes/no (✓/✗).

| | **Graph-Based** | | **Quantization-Based** | | | **QG** | **PG** | **PAG** | |
|---|---|---|---|---|---|---|---|---|---|
| **Criteria** | HNSW | Vamana | IVFPQFS[1] | ScaNN[1] | RaBitQ+ | SymQG | HNSW+KS2 | PAG-Base | PAG-Lite |
| **D1.** QPS-recall | Tier-2 | Tier-2 | Tier-1 | Tier-1 | Tier-1 | Tier-3 | Tier-3 | **Tier-4** | **Tier-3** |
| **D2.** Indexing time | Tier-1 | Tier-1 | Tier-3 | Tier-3 | Tier-3 | Tier-2 | Tier-1 | **Tier-2** | **Tier-4** |
| **D3.** Memory footprint[2] | Tier-3 | Tier-3 | Tier-4 | Tier-4 | Tier-4 | Tier-1 | Tier-2 | **Tier-3** | **Tier-4** |
| **D4.** High-dim. scalability | ✓ | ✓ | ✓ | ✓ | ✓ | ✗ | ✓ | ✓ | ✓ |
| **D5.** Retrieval size robustness | ✓ | ✓ | ✓ | ✓ | ✓ | ✗ | ✓ | ✓ | ✓ |
| **D6.** Online insertion support[3] | ✓ | ✓ | ✓ | ✓ | ✓ | ✗ | ✗ | ✓ | ✓ |

[1] Reranking is enabled for IVFPQFS and ScaNN.
[2] Memory footprint for searching, rather than indexing, is compared.
[3] For any streaming workload $Q$ composed of search and insertion queries, we say an algorithm supports online insertion, if the amortized cost of processing a query in $Q$ is $O(1)$ times the cost of a search. In other words, the index can be incrementally updated with minimal cost. See Sec. 4.2 for the analysis of PAG.

*Graph-Based Methods.* These methods construct a similarity graph connecting nearby vectors and hop towards the neighborhood of $q$. Notable methods are HNSW (Malkov & Yashunin, 2020), NSG (Fu et al., 2019), and DiskANN (Subramanya et al., 2019). They are generally competitive in QPS-recall but slow in building indexes.

*Quantization-Based Methods.* These methods compress vectors and rank them using approximate (i.e., quantized) distances to $q$ in pursuit of efficiency. Representative methods are IVFPQ (Jégou et al., 2011) and ScaNN (Guo et al., 2020a). They are fast in building indexes and save memory, but their QPS-recall is generally inferior to graph-based methods.

*Projection-Based Methods.* These methods use random (e.g., E2LSH (Andoni & Indyk, 2005), Falconn (Andoni et al., 2015), and CEOs (Pham, 2021)) or data-dependent (e.g., learning to hash (Wang et al., 2018)) projections of the original vectors for indexing. Although many of them enjoy theoretical guarantees, they are less competitive than graph- and quantization-based methods in QPS-recall performance, hence becoming less popular in modern top-$K$ ANNS.

*Tree-Based Methods.* They utilize tree indexes. Notable methods include k-d tree (Bentley, 1990), cover tree (Beygelzimer et al., 2006), and Annoy (Bernhardsson, 2013). Like projection-based methods, they are less widely used for modern ANNS due to limited QPS-recall performance.

As discussed above, graph-based methods are good at QPS-recall. But they compute *exact* distances between vectors, which is computationally costly. To address this weakness, recent advancements attempt to integrate other techniques, in particular, quantization or projection, into graphs.

*Quantized Graph (QG) Methods.* They construct a similarity graph based on quantized vectors instead of the original ones, thereby replacing exact distances with *approximate* values. Representative methods are NGT-QG (Yahoo! Japan, 2023),

LVQ (Aguerrebere et al., 2023), and SymphonyQG (Gou et al., 2025a). QG methods can achieve very high QPS-recall performance, yet they are sensitive to the data distribution and many of them do not perform very well in D3 – D6.

*Projection + Graph (PG) Methods.* Unlike QG, they operate on original vectors and reduce unnecessary exact distance computations. To this end, they employ projection techniques to test whether a neighbor needs to be explored (called routing tests). Representative methods are FINGER (Chen et al., 2023), PEOs (Lu et al., 2024), and KS2 (Lu et al., 2025). Despite QPS-recall improvement on top of graph-based methods, such improvement comes at the cost of indexing time, memory footprint, and support for online insertions.

Table 1 compares notable methods implemented in leading vector databases and recent ones representing state of the art.

## 1.2. Our Solution

We propose a new ANNS framework, **P**rojection-**A**ugmented **G**raph (PAG), which integrates projection into a similarity graph and achieves superior performance for all the six criteria. Unlike PG methods, PAG treats projection as a fundamental building block of graph construction rather than as a plug-in. The rationale of PAG is to accommodate both exact and approximate distances within a *unified* framework, addressing two key issues: (1) *when distances shall be computed exactly and when they shall be approximated*, and (2) *how exact and approximate distances are compared during indexing and searching to enhance the performance.*

PAG significantly reduces the computational cost of searching (D1) and indexing (D2) by carefully determining whether exact distance computation is needed. Such procedure relies on asymmetric comparisons between exact and approximate distance values obtained from space-efficient and adjustable random-projection structures (D3), as opposed to symmetric comparisons in QG. Moreover, the asymmetric distance

comparisons, as guided by random-projection-based statistical tests, only need to tell which value is larger, thereby achieving accuracy while preserving efficiency. Meanwhile, PAG inherits graph-based methods' scalability to high dimensionality (D4) and robustness against retrieval size (D5), and follows the search-and-insertion paradigm (e.g., HNSW) to accommodate online insertions (D6).

Our technical contributions are summarized as follows.

(1) Following the idea of routing tests in PEOs (Lu et al., 2024) and KS2 (Lu et al., 2025), we derive a Probabilistic Routing Test (PRT) function (Sec. 3.2). The theoretical result (Theorem 3.1), along with Lemma 4.3 in Lu et al. (2025), provides a complete theoretical explanation of PRT. In addition, our work is the first to apply PRT to graph construction.

(2) We propose a data structure called Test Feedback Buffer (TFB) (Sec. 3.3), which refines the threshold setting in PRT and enables the reuse of false positives generated by PRT. By incorporating TFB into PRT, we obtain the PRT-TFB test as a core technique for accelerating both indexing and searching.

(3) We propose a statistical test called Probabilistic Edge Selection (PES) (Sec. 3.4), which is derived from Theorem 3.1 and can expand in-degrees when necessary. In collaboration with PRT, PRT-PES can improve the search performance for hard datasets on which traditional graph indexes perform poorly, while incurring very small indexing overhead.

(4) PRT, TFB, and PES constitute PAG (Fig. 1). We show how the three components interact within PAG, with implementation details and complexity analysis provided.

(5) We conduct experiments on six modern (post-2023) datasets covering text, image, and multimodal data, with dimensionality ranging from 384 to 3072 and retrieval size from 10 to 1000. The results show that PAG achieves the best QPS-recall performance (up to $5\times$ faster than HNSW), and its superiority is particularly evident on datasets with higher dimensionality. PAG remains dominant as $K$ increases. By adjusting parameters, PAG can deliver the fastest indexing speed and lowest memory footprint on most datasets while maintaining competitive QPS-recall. PAG also performs well on the datasets in the ANN-Benchmarks, showcasing its compatibility with data generated by legacy models.

## 2. Problem Setting

Since PAG adopts a search-and-insertion paradigm, its search process, similar to HNSW, can be seen as a special case of index construction (i.e., no update to the graph), and we can focus on the problems that arise in index construction. Because constructing a similarity graph is essentially determining its edge set, we consider the following question: *Can we design statistical methods that select edges efficiently (i.e., fast indexing) and effectively (i.e., fast searching)?* To answer this question, we first review the existing edge-selection strategy, and then formalize the problems to be solved.

### 2.1. Background of RobustPrune

Given a directed graph, let $v$ be a node whose edges are to be determined. To identify its in-neighbor set $N_{\text{in}}(v)$ and out-neighbor set $N_{\text{out}}(v)$, state-of-the-art methods, such as HNSW, NSG and Vamana, employ a strategy known as RobustPrune to decide whether a candidate edge should be preserved. Take the construction of $N_{\text{in}}(v)$ as an example. For a candidate neighbor $u$ of node $v$, the pruning criterion considers the following set [2]:

$$S_{u,v} = \{w \in N_{\text{out}}(u) \mid \|w - v\| \le \|v - u\|\}. \quad (1)$$

In RobustPrune, if there exists $w$ in $S_{u,v}$ such that $\|w - u\| \le \|v - u\|$, $u$ is not added to $N_{\text{in}}(v)$; otherwise, $N_{\text{in}}(v)$ is updated as $N_{\text{in}}(v) \cup \{u\}$. This criterion admits an intuitive interpretation: if $\|w - u\| \le \|v - u\|$ holds for some $w$, it is likely that one can reach $v$ from $u$ via $w$ along a monotonic search path (Fu et al., 2019), implying that the direct edge $\vec{uv}$ is redundant. Similarly, $N_{\text{out}}(v)$ can be constructed by RobustPrune with the roles of $u$ and $v$ reversed. In sum, existing graph-based methods determine $N_{\text{out}}(v)$ and $N_{\text{in}}(v)$ sequentially in the following manner.

$\underline{N_{\text{out}}(v)}$. We take $v$ as the query and conduct an ANNS from $v$ to obtain a candidate set, which is stored in a priority queue $P$. RobustPrune is then applied to $P$ to determine $N_{\text{out}}(v)$.

$\underline{N_{\text{in}}(v)}$. We check every $u$ in $N_{\text{out}}(v)$ and preserve those $u$'s retained by RobustPrune as $N_{\text{in}}(v)$.

### 2.2. Towards Fast Graph Construction

During graph construction, most of the time is spent on ANNS for the candidate set of $N_{\text{out}}(v)$. Although we can reduce the graph construction parameter (e.g., $efC$ in HNSW) for fast graph construction, the query processing performance degrades accordingly. Thus, we raise the following question.

**Q1.** *How can we accelerate the ANNS for $v$ while preserving the query processing performance?*

Prior studies on PEOs and KS2 have shown that PRT has the potential to be an appropriate answer to Q1. For each visited node $u$ and its out-neighbor $w$, PRT sends $(u, v, w)$ to a probabilistic test function to see whether they can pass the test, and then determines whether the exact distance between $v$ and $w$ needs to be computed based on the test result. Consequently, PRT avoids unnecessary distance computations, leading to faster search. In KS2 (Lu et al., 2025), it is shown that the routing test under $\ell_2$ and cosine distances is equivalent to the angle-thresholding problem when vector norms

---

[2]For Vamana, we assume its pruning ratio $\alpha = 1$ and omit it in the inequality.

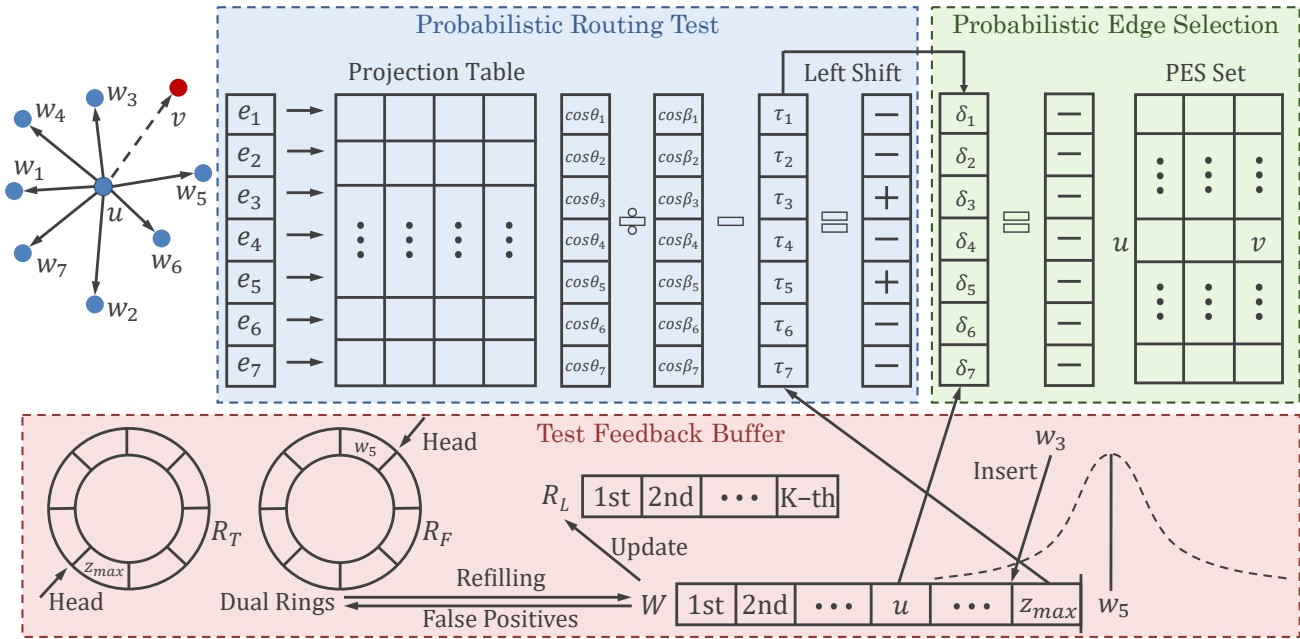

*Figure 1.* An overview of PAG. In this example, $u$ has 7 out-neighbors $\{w_i\}_{i=1}^7$. Let $\{e_i\}_{i=1}^7$ denote the edges between $u$ and $\{w_i\}_{i=1}^7$, which are sent to PRT, where the threshold is determined by $z_{\mathbf{max}}$. As a result, only $w_3$ and $w_5$ pass PRT, and their exact distances to $v$ are computed. By distance comparison, node $w_5$ is identified as a false positive, not added to $W$, and thus sent to $R_F$. Node $w_3$ is inserted into $W$, causing $z_{\mathbf{max}}$ to be ejected from $W$ and sent to $R_T$. The two rings $R_F$ and $R_T$ are merged and refilled into $W$. On the other route, by a left shift, the signs of both $w_3$ and $w_5$ are reversed, and all signs become negative, indicating that no witness is detected by PES. Consequently, $\vec{uv}$ is treated as a candidate edge and added to the PES set.

are stored explicitly. Therefore, PRT can be formulated as follows:

**Problem 2.1.** (**Probabilistic Routing Test**) Given $u, v, w \in \mathbb{S}^{d-1}$ and a threshold $-1 \leq \tau \leq 1$, construct a random-projection vector set $\mathcal{F}$ and design a test function w.r.t $\mathcal{F}$, i.e., PRT : $(u, v, w, \tau) \to RV$ with time complexity $o(d)$, where $RV$ denotes the set of one-dimensional random variables, such that if $\cos(\angle(v - u, w - u)) \geq \tau$, then $\mathbb{P}[\text{PRT}(u, v, w, \tau) \geq 0] \geq 0.5$. Otherwise, $\mathbb{P}[\text{PRT}(u, v, w, \tau) < 0] \to 1$ as $|\mathcal{F}| \to \infty$.

Different from the routing test functions used in PEOs and KS2, we take $\tau$ as an explicit input. In this paper, we use another method to determine $\tau$ in order to further improve the efficiency of the routing test.

### 2.3. Towards High Graph Connectivity

From the process of determining $N_{\text{in}}(v)$, we observe that its candidate set is restricted to $N_{\text{out}}(v)$. Recent work (Wang et al., 2025a) shows that $|N_{\text{out}}(v)|$ may be too small to reliably determine incoming edges, causing some nodes to have very small in-degrees. Such nodes may become unreachable during search, which partly explains why HNSW performs poorly on certain real-world datasets. On the other hand, for each checked node $u$, the worst-case time complexity of RobustPrune can reach $O(|N_{\text{out}}(u)|d)$, implying that a straightforward enlargement of the candidate set for

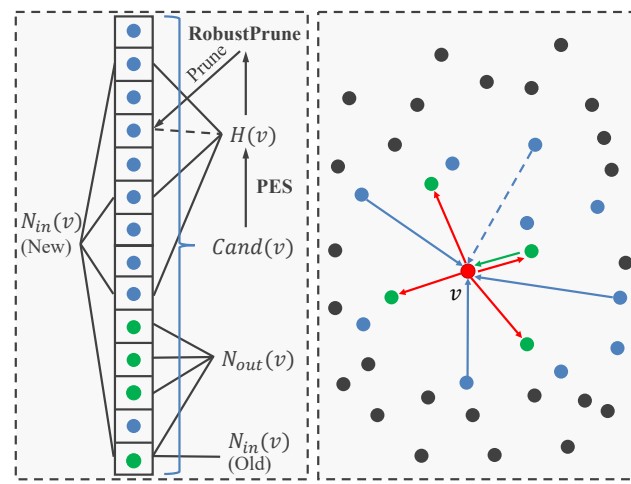

*Figure 2.* Illustration of PES (left: the role of PES; right: geometric illustration). Let $v$ be the node to be inserted. The four green nodes are out-neighbors of $v$. Without PES, we can obtain only a single in-neighbor via RobustPrune. By taking all other visited nodes (the blue ones) into account, we apply PES, followed by RobustPrune. As such, we can identify three additional promising in-neighbors, thereby strengthening the connectivity of the neighborhood of $v$.

determining incoming edges is expensive. Considering this dilemma, we raise the following question.

Q2. *How can we efficiently detect incoming edges outside $N_{\text{out}}(v)$ that are useful for improving query processing performance but hard to be identified by RobustPrune?*

To answer this question, we formulate a problem as follows.

**Problem 2.2.** (**Probabilistic Edge Selection**) Given $\boldsymbol{u}, \boldsymbol{v} \in \mathbb{S}^{d-1}$, construct a random-projection-vector set $\mathcal{F}$ and design a probabilistic edge selection function PES : $(\boldsymbol{u}, \boldsymbol{v}) \to RV$ whose time complexity is $O(|N_{\text{out}}(\boldsymbol{u})|)$, such that if $S_{\boldsymbol{u},\boldsymbol{v}}$ is non-empty, then $\mathbb{P}[\text{PES}(\boldsymbol{u}, \boldsymbol{v}) \geq 0] \geq 0.5$. Otherwise, $\mathbb{P}[\text{PES}(\boldsymbol{u}, \boldsymbol{v}) < 0] \to 1$ as $|\mathcal{F}| \to \infty$.

We emphasize that the PES test is far more than a fast probabilistic implementation of RobustPrune. This is because the PES test can be applied to all visited nodes during the ANNS of $\boldsymbol{v}$, whose number is typically much larger than $N_{\text{out}}(\boldsymbol{v})$ in practice, leading to a better graph index in terms of connectivity (Fig. 2).

# 3. Projection-Augmented Graph

We show that Problems 2.1 and 2.2 can be solved in a unified framework PAG, which contains three components: Probabilistic Routing Test (PRT), Test Feedback Buffer (TFB), and Probabilistic Edge Selection (PES).

## 3.1. Neighborhood Relations via Random Projection

We present an asymptotic result that characterizes the relationship between multiple angles in high-dimensional spaces and their corresponding projection values onto a certain projection vector. This result forms the theoretical basis for all the three components. We first describe how to construct the random-projection vector set $\mathcal{F}$ that appears in Problems 2.1 and 2.2, following an approach similar to that used in KS2 (Lu et al., 2025). Specifically, we apply a fixed random orthogonal transform to all data and then divide the original space $\mathbb{R}^d$ into $L$ subspaces, each of dimension $d/L$. In each subspace, we generate multiple cross-polytopes and apply an independent rotation to each, producing a total of $m$ unit vectors on $\mathbb{S}^{d/L-1}$, which are then scaled by $1/\sqrt{L}$. By concatenating one scaled vector from each subspace, we obtain a total of $m^L$ normalized vectors $\{\boldsymbol{r}_j\}_{j=1}^{m^L}$, which together form the set $\mathcal{F}$. Consider $\boldsymbol{v}$ as the node to be inserted and $\boldsymbol{u}$ as the candidate neighbor to be checked, let $N_{\text{out}}(\boldsymbol{u}) := \{\boldsymbol{w}_i\}_{i=1}^t$. We define $\{\alpha_i\}_{i=1}^t$ as follows.

$$\alpha_i := \arccos \frac{\langle \boldsymbol{w_i} - \boldsymbol{u}, \boldsymbol{v} - \boldsymbol{u} \rangle}{\|\boldsymbol{w_i} - \boldsymbol{u}\| \|\boldsymbol{v} - \boldsymbol{u}\|}, \quad 1 \leq i \leq t \quad (2)$$

Since $\|\boldsymbol{w_i} - \boldsymbol{u}\|$ and $\|\boldsymbol{v} - \boldsymbol{u}\|$ can be pre-computed, we aim to estimate $[\cos \alpha_1, \ldots, \cos \alpha_t]^\top$ without exact inner product computation in Eq. (2). We will show that this can be realized by $\mathcal{F}$. For each $\boldsymbol{w_i}$ ($1 \leq i \leq t$), let $\boldsymbol{r_i^*} \in \mathcal{F}$ be the reference vector that has the smallest angle with $\boldsymbol{w_i} - \boldsymbol{u}$, and denote this smallest angle by $\beta_i$. We use $\cos \theta_i$ to denote the cosine of the angle between $\boldsymbol{r_i^*}$ and $\boldsymbol{v} - \boldsymbol{u}$, and subscript $l$ to denote the $l$-th sub-vector of the original vector in $\mathbb{R}^d$, $1 \leq l \leq L$. For simplicity, write $w_{il} := (\boldsymbol{w_i} - \boldsymbol{u})_l$ and $v_l := (\boldsymbol{v} - \boldsymbol{u})_l$.

We introduce the following assumptions for each $i$:

(A1) $\|(\boldsymbol{w_i} - \boldsymbol{u})\| = 1$ and $\|(\boldsymbol{v} - \boldsymbol{u})\| = 1$.

(A2) $\eta_L := \max_{i,l} \left\{ \begin{array}{l} \left| \sqrt{L} \|w_{il}\| - 1 \right|, \\ \left| \sqrt{L} \|v_l\| - 1 \right|, \\ |L\langle v_l, w_{il} \rangle - \cos \alpha_i| \end{array} \right\} \to 0.$

Here, A1 does not lose generality, and A2 is mild for large $d$ and $L$ (see the remarks in Appendix B). Then, the following theorem shows that $\{\alpha_i\}_{i=1}^t$ can be asymptotically estimated by $\{\beta_i\}_{i=1}^t$ and $\{\theta_i\}_{i=1}^t$.

**Theorem 3.1.** *Under A1, A2 and the residual regularity assumptions stated in Appendix B, for fixed $t$, as $L \to \infty$, $d/L \to \infty$, and $m$ grows sufficiently fast with respect to $d/L$, for $\mathbf{Y} = [\cos \beta_1, \ldots, \cos \beta_t]^\top$ in its typical set, $\mathbf{X} = [\cos \theta_1, \ldots, \cos \theta_t]^\top$, conditioned on $\{\alpha_i, \beta_i\}_{i=1}^t$, is asymptotically Gaussian:*

$$\mathcal{L}\left(\mathbf{X} \mid \{\alpha_i, \beta_i\}_{i=1}^t\right) = \mathcal{N}(\boldsymbol{\mu}_{\alpha,\beta} + \boldsymbol{r}_L, \bar{\Sigma}_{m,L}) + o(1) \quad (3)$$

*where $\boldsymbol{\mu}_{\alpha,\beta} = [\cos \alpha_1 \cos \beta_1, \ldots, \cos \alpha_t \cos \beta_t]^\top$, $\|\boldsymbol{r}_L\| = O(\eta_L)$, and $\|\bar{\Sigma}_{m,L}\| = O(\omega_{m,L}/L)$. Here, $\omega_{m,L} := \eta_L + \epsilon_m$, where $\epsilon_m$ is the one-block projection error defined in Appendix B. $\omega_{m,L}$ is bounded and tends to zero when $\eta_L \to 0$ and $m$ grows sufficiently fast with respect to $d/L$.*

**Remarks.** (1) Theorem 3.1 establishes a geometric relationship among multiple out-neighbors of $u$ and $v$, allowing us to estimate the relative location of $v$ w.r.t. $u$ within the neighborhood of $u$. This relationship is the key to both PRT and PES. (2) In our proof, the block decomposition shows that the projection statistic satisfies $\cos \theta_i = \cos \alpha_i \cos \beta_i$ plus a small residual term. The parameter $L$ controls the aggregation error through the $1/L$ factor in the covariance, while $m$ controls the one-block projection error. In practice, moderate values of $m$ and $L$ are sufficient. Weak results under finite $(m, L)$ can be found in Lemma 4.3 in the KS2 paper (Lu et al., 2025). (3) $\mathbf{X}$ can be computed efficiently by AVX512 on modern CPUs. Vector $[\cos \beta_1, \ldots, \cos \beta_t]^\top$ can be pre-computed, meaning that we can obtain an efficient way to estimate all $\cos \alpha_i$'s simultaneously.

## 3.2. Probabilistic Routing Test

Let $\tau_i$ be a threshold w.r.t $\boldsymbol{w_i}$. Our PRT function is as follows:

$$\text{PRT}(\boldsymbol{u}, \boldsymbol{v}, \boldsymbol{w_i}, \tau_i) = \frac{\cos \theta_i}{\cos \beta_i} - \tau_i. \quad (4)$$

For fixed $(\boldsymbol{u}, \boldsymbol{v})$, if the value of PRT is positive, the corresponding $\boldsymbol{w_i}$ passes the PRT. Based on Theorem 3.1, under asymptotic assumptions and away from the decision boundary, the PRT function provides an asymptotically valid approximation to the probabilistic test required in Problem 2.1. Here, the setting of $\tau_i$ will be postponed to Sec. 3.3.

**Remarks.** Following the use of routing test in PEOs and KS2, exact distances are computed only between $\boldsymbol{v}$ and those $\boldsymbol{w_i}$'s that pass PRT (see Fig. 1). The PRT function has the same structure as the KS2 test function (Lu et al., 2025), except for the setting of the threshold $\tau$. However, since the PRT and KS2 tests are derived using different principles, their theoretical results are complementary. Lemma 4.3 in the KS2 paper shows that, for every single $\boldsymbol{w_i}$, even for finite $(m, L)$, if $\cos(\angle(\boldsymbol{v} - \boldsymbol{u}, \boldsymbol{w_i} - \boldsymbol{u})) \geq \tau_i$, then $\mathbb{P}[\text{PRT}(\boldsymbol{u}, \boldsymbol{v}, \boldsymbol{w_i}, \tau_i) \geq 0] \geq 0.5$ still holds. Our result, on the other hand, characterizes the concrete asymptotic distribution and reveals the impact of $L$ on the covariance, thereby explaining how $L$ is used to adjust the estimation accuracy.

### 3.3. Test Feedback Buffer

For the PRT function, a key issue is how to set an appropriate threshold $\tau$. In PEOs and KS2, $\tau$ is simply determined by the current furthest point in the result list (priority queue) $P$. However, due to the existence of false positives (FPs) generated by PRT, some points may pass PRT but cannot be added to $P$, which implies that the exact distance computations w.r.t such points are redundant. On the other hand, the Gaussian distribution established in Theorem 3.1 implies that, with high probability, the actual distances of these FPs to the query $\boldsymbol{q}$ are not much larger than the threshold $\tau$. This observation naturally raises the following question: *can we incrementally increase the threshold $\tau$ so that the currently generated FPs can be reused in subsequent search processes?*

We give an affirmative answer to this question and propose TFB, which consists of four components: the result/candidate list $R_L$, the working set $W$, and two ring buffers $R_F$ and $R_T$, where $|W| = |R_F| = |R_T|$. The search procedure is divided into multiple rounds. In the $j$-th round, we operate on $W$ and adopt the standard Best-First Search (BFS) strategy to update its nodes. During the BFS over $W$, in addition to the nodes stored in $W$, there are two types of nodes whose exact distances to the query $\boldsymbol{q}$ are computed. The first type consists of nodes ejected from $W$; these nodes are inserted into $R_T$. The second type consists of false-positive (FP) nodes that pass the PRT but cannot be added to $W$; these nodes are inserted into $R_F$. After all nodes in $W$ have been visited, we update $R_L$ and clear $W$. We then merge $R_F$ and $R_T$ and sort the combined list by distance to $\boldsymbol{q}$. The sorted nodes are first inserted into $W$ until it is full, and the remaining nodes are inserted into $R_T$. Finally, $R_F$ is cleared. The ANNS is completed after several such rounds. In the $j$-th round, let $\boldsymbol{z}_{\max}$ denote the furthest node in $W$. $\tau_i$ in Eq. (4) is set to be the threshold on the cosine of the angle between $\boldsymbol{w_i} - \boldsymbol{u}$ and $\boldsymbol{v} - \boldsymbol{u}$ required for $\boldsymbol{w_i}$ to be admitted into $W$, i.e.,

$$\tau_i = \frac{\|\boldsymbol{u} - \boldsymbol{w_i}\|^2 + \|\boldsymbol{u} - \boldsymbol{v}\|^2 - \|\boldsymbol{z}_{\max} - \boldsymbol{v}\|^2}{2\|\boldsymbol{v} - \boldsymbol{u}\|\|\boldsymbol{w_i} - \boldsymbol{u}\|}. \quad (5)$$

*PRT-TFB Test.* We call the combination of Eq. (4) and Eq. (5)

the PRT-TFB test. Compared with existing routing test methods such as PEOs and KS2, which operate on the whole priority queue $P$, the PRT-TFB test has the following two advantages. (1) For large $efC$ or large $efS$, which denotes the maximum number of working-set nodes visited during search, the size of $W$ is much smaller than that of $P$, and the movement of elements in $W$ is much faster than that in $P$. (2) Each FP has a good chance of being selected as the visited node in the future rounds only if it remains in either of two rings, ensuring that its exact distance to $\boldsymbol{q}$ can be utilized at a certain time point.

### 3.4. Probabilistic Edge Selection

In RobustPrune, $\|\boldsymbol{w_i} - \boldsymbol{v}\|$ needs to be smaller than $\|\boldsymbol{v} - \boldsymbol{u}\|$ for $\boldsymbol{w_i}$ to enter $S_{u,v}$ and serve as a potential witness for pruning $\boldsymbol{u}$. Let $\delta_i := \|\boldsymbol{w_i} - \boldsymbol{u}\|/(2\|\boldsymbol{v} - \boldsymbol{u}\|)$ denote the threshold on the cosine of the angle between $\boldsymbol{w_i} - \boldsymbol{u}$ and $\boldsymbol{v} - \boldsymbol{u}$, such that $\|\boldsymbol{w_i} - \boldsymbol{v}\| < \|\boldsymbol{v} - \boldsymbol{u}\|$. Then, the PES function is designed as follows.

$$\text{PES}(\boldsymbol{u}, \boldsymbol{v}) = \max_{1 \leq i \leq t} \left( \frac{\cos \theta_i}{\cos \beta_i} - \delta_i \right). \quad (6)$$

By Theorem 3.1, under asymptotic assumptions and away from the decision boundary, the PES function in Eq. (6) provides an asymptotically valid approximation to the probabilistic properties in Problem 2.2. In practice, if the value in Eq. (6) is negative, we say that $\overrightarrow{uv}$ survives the PES screening, and this edge will be regarded as a promising candidate for further examination by RobustPrune.

*PRT-PES Collaboration.* Clearly, the difference between the two thresholds, that is, $\tau_i - \delta_i$, does not require computing $\|\boldsymbol{v} - \boldsymbol{w_i}\|$ and can be obtained from quantities already available during the PRT-TFB test. Thus, while scanning the out-neighbors of $u$ for PRT, we can update the PES statistic with only $O(1)$ additional work per checked neighbor.

*PES Set.* For every $u$ whose PES value is negative, we do not check $\overrightarrow{uv}$ by RobustPrune immediately. Instead, we add $\overrightarrow{uv}$ into a so-called PES set for later examination (see Fig. 1).

## 4. Implementation and Analysis

### 4.1. Implementation

*Indexing Phase.* PAG inserts every $\boldsymbol{v} \in \mathcal{D}$ sequentially into the graph. The indexing procedure consists of two per-node steps and one batch-level PES refinement step.

**Step 1.** PAG performs ANNS equipped with the PRT-TFB test to obtain the elements stored in $R_L$, and then uses RobustPrune to determine $N_{\text{out}}(\boldsymbol{v})$ and $N_{\text{in}}(\boldsymbol{v})$.

**Step 2.** Let $\text{Cand}(\boldsymbol{v})$ denote the set of all nodes visited during ANNS. PAG uses PRT-PES to select additional candidate incoming edges from $\text{Cand}(\boldsymbol{v})$, and stores them in the

bounded PES candidate buffer $H(v)$.

Based on our preceding discussion, PRT-TFB ensures the efficiency of ANNS, while PRT-PES detects more promising edges. The stored candidates are then verified in Step 3.

**Step 3.** The final incoming edges are determined by applying RobustPrune to those stored candidates.

*Searching Phase.* The query processing with PAG is exactly the PRT-TFB-based ANNS for a query $q$.

*Online Insertion.* Like HNSW, PAG naturally supports on-line insertion. The PES set is checked only after enough new nodes have been inserted into the graph.

### 4.2. Complexity Analysis

*Time complexity.* Searching with HNSW takes $O(n'Md)$ time, where $n'$ is the number of visited nodes. Although a rigorous analysis of $n'$ remains an open problem, $n'$ can be roughly estimated as $O(M \log n)$, where $2M$ is the maximum out-degree. In contrast, the complexity of PAG is $O(n'ML + \gamma Mn'd)$, where $L \ll d$ is the user-specified parameter and $\gamma$ denotes the ratio of nodes that passed the projection-based test relative to the total number of candidates checked. $\gamma$ is small in practice (generally below 0.2).

As for indexing, due to the search-and-insertion paradigm, we focus on the insertion of $v$, for which the complexity is $O(n'ML + \gamma Mn'd + efC \cdot Md + Mdm)$. Here, $O(efC \cdot Md)$ is the complexity of node connection, and $O(Mdm)$ is used to compute projection information.

*Space complexity.* The space complexity of searching with PAG is $O(nd + nM + nML)$, where $O(nd)$ corresponds to the dataset, $O(nM)$ corresponds to the edge set and $O(nML)$ corresponds to the inference structure. In the indexing phase, we only require an additional space of at most $O(Mn)$ to store the edges in the PES set because PAG keeps at most $2M$ PES candidates in $H(v)$ for each node $v$. To reduce the memory footprint, we can follow LVQ (Aguerrebere et al., 2023) and represent floats with 2 bytes.

*Online Insertion.* The PES refinement is processed after an insertion batch. Since PAG keeps at most $2M$ PES candidates in $H(v)$ for each inserted node $v$, a batch of $B$ insertions contains $O(BM)$ deferred candidates. Verifying them costs $O(BM^2d)$, i.e., $O(M^2d)$ amortized per inserted node.

## 5. Experiments

### 5.1. Experimental Setup

*Data Statistics.* In the main experiments, we evaluate PAG on eight datasets: DBpedia1536 (Qdrant, 2024a) (a.k.a., OpenAI-1536), DBpedia3072 (Qdrant, 2024b) (a.k.a., OpenAI-3072), WoltFood (Qdrant, 2024c), Ama-

zonBooks (Kang et al., 2024), DataCompDr (Apple, 2025), MajorTOM (Major TOM, 2024), GloVe (Pennington et al., 2014), and DEEP100M (Simhadri et al., 2021). The first six are modern text, image, and multimodal (text-to-image) datasets generated by recent embedding models. The latter two are widely-used legacy datasets (downloaded from The Similarity Search Team, CUHK (2018)). We also report the experimental results on other legacy datasets (Word2Vec, GIST, ImageNet, and SIFT10M) in Appendix G.3. Dataset statistics are reported in Table 2. The queries of DataCompDr are out-of-distribution (OOD). Fig. 8 in Appendix F plots the data and query distributions of this dataset.

*Baselines.* Baselines include HNSW (Malkov & Yashunin, 2020), Vamana (Subramanya et al., 2019), SymQG (Gou et al., 2025a), ScaNN (Guo et al., 2020a), IVFPQFS (Jégou et al., 2011), and RaBitQ+ (Gao et al., 2025). We design two variants of PAG by tuning its parameters, as shown in Table 4.

### 5.2. Experimental Results

We set the default retrieval size $K = 100$.

*D1. QPS-Recall.* Fig. 3 plots the QPS-recall performance. PAG-Base performs the best on all the modern datasets except for high recall settings on WoltFood. Its speedup over HNSW can be up to 5 times. For legacy datasets, PAG-Base is the best on GloVe and only second to SymQG on DEEP100M. PAG-Lite also delivers competitive search performance and is the runner-up on DBpedia1536, DBpedia3072, and MajorTOM.

*D2. Indexing Time.* Fig. 4 (left) shows the indexing time. PAG-Base requires only 20–40% of the indexing time of HNSW, and is faster than SymQG in most cases. PAG-Lite's indexing time is comparable to quantization-based methods, and is further reduced to $0.5\times$ on high-dimensional datasets, where it attains the lowest indexing time.

*D3. Memory Footprint.* Fig. 4 shows the memory usage in indexing (middle) and searching (right) phases. PAG-Base uses more memory than HNSW on lower-dimensional datasets, but the difference is negligible on higher-dimensional datasets. PAG-Base consistently uses much less memory than SymQG. PAG-Lite achieves the smallest memory footprint in 4 out of 8 cases for both indexing and searching. Usage breakdown is reported in Appendix G.4.

*D4. High-dimensional Scalability.* The competitiveness of PAG-Base and PAG-Lite over $d \in [96, 3072]$, as shown in Figs. 3 and 4, is consistent, showcasing its insensitivity to dimensionality. In addition, the advantage of PAG in QPS-recall becomes more pronounced on high-dimensional datasets. In contrast, SymQG reports very low recall on high-dimensional datasets DBpedia1536, DBpedia3072, and DataCompDr, where $d \in \{1536, 3072\}$ [3].

---

[3]This is due to the use of unsigned 16-bit integers for parallelism

*Table 2.* Dataset statistics.

| Name/Source | Dataset Size | Query Size | Dim. | OOD | Type | Embedding Model | Distance Measure |
|---|---|---|---|---|---|---|---|
| **Modern Datasets** | | | | | | | |
| DBpedia1536 | 999,000 | 1,000 | 1,536 | No | Text | text-embedding-3-large (OpenAI, 2024) | Euclidean |
| DBpedia3072 | 999,000 | 1,000 | 3,072 | No | Text | text-embedding-3-large (OpenAI, 2024) | Euclidean |
| WoltFood | 1,719,611 | 1,000 | 512 | No | Image | clip-ViT-B-32 (Radford et al., 2021) | Euclidean |
| DataCompDr | 12,779,520 | 1,000 | 1,536 | Yes | Text-to-Image | coca_ViT-L-14 (Yu et al., 2022) | Euclidean |
| AmazonBooks | 15,928,208 | 1,000 | 384 | No | Text | all-MiniLM-L12-v2 (Wang et al., 2020) | Euclidean |
| MajorTOM | 56,506,400 | 10,000 | 1,024 | No | Image | DINOv2 (Oquab et al., 2024) | Euclidean |
| **Legacy Datasets** | | | | | | | |
| Word2Vec | 1,000,000 | 1,000 | 300 | No | Text | Word2Vec (Mikolov et al., 2013) | Euclidean |
| GIST | 1,000,000 | 1,000 | 960 | No | Image | GIST (Oliva & Torralba, 2001) | Euclidean |
| GloVe | 1,193,514 | 1,000 | 200 | No | Text | GloVe (Pennington et al., 2014) | Euclidean |
| ImageNet | 2,340,373 | 200 | 150 | No | Image | dense SIFT (Lazebnik et al., 2006) | Euclidean |
| SIFT10M | 10,000,000 | 1,000 | 128 | No | Image | SIFT (Lowe, 2004) | Cosine |
| DEEP100M | 100,000,000 | 1,000 | 96 | No | Image | GoogLeNet (Szegedy et al., 2015) | Cosine |

*Figure 3.* QPS-recall, $K = 100$. SymQG runs out of memory on MajorTOM. Recall is plotted on a logarithmic scale.

**D5. Retrieval Size Robustness.** Besides the QPS-recall for $K = 100$ in Fig. 3, we report the results for $K = 10$ in Fig. 13 and $K = 1000$ in Fig. 14. PAG methods are highly competitive across the three $K$ values. When $K = 10$, PAG-Base achieves comparable performance and even outperforms SymQG in the high-recall ($\geq 95\%$) region on all datasets except DEEP100M. When $K = 1000$, PAG-Base remains the best, while the performance of SymQG degrades significantly. To highlight the performance comparison, we also plot in Fig. 15 the QPS-recall by varying $K$ from 200 to 800. It can be seen that the gap between PAG-Base and SymQG grows when $K$ moves towards larger values. This observation showcases the robustness of PAG in various applications that differ in retrieval size.

**D6. Online Insertion Support.** To evaluate the query processing performance with online insertions, we consider the following workload. We randomly sample from the corpus 10,000 vectors as insertion queries and another 10,000 vectors as search queries. The 20,000 vectors are divided into

20 batches, each with 1,000 vectors. The insertion batches and the search batches are interleaved as a workload, with insertion as the first batch. The rest of the corpus is used to build the initial index. Note that DataCompDr is not OOD in this setting because its original query set (in Table 2) is not used. To make the processing times of insertion queries and search queries on the same scale, we set $efS = efC$ and tune these two parameters to control the recall. Fig. 5 compares PAG-Base and HNSW, plotting their QPS-recall performances of insertion and search. Insertion queries are slightly slower to process than search queries for both methods. PAG-Base is much faster than HNSW in both insertion and search speeds, and PAG-Base's insertion is even faster than HNSW's search. Similar to PAG-Base's advantage in search, its speedup over HNSW in insertion can be up to 5 times, demonstrating its efficiency in online insertions.

**Ablation Study.** We choose four datasets, DBpedia3072, WoltFood, AmazonBooks, and DataCompDr, for ablation study. Among the six modern datasets, they cover the lowest and highest dimensionality, text and images, as well as OOD. Figs. 6 and 7 show the effectiveness of PAG's components.

in its source code, causing an overflow under high dimensions. Fixing it is non-trivial.

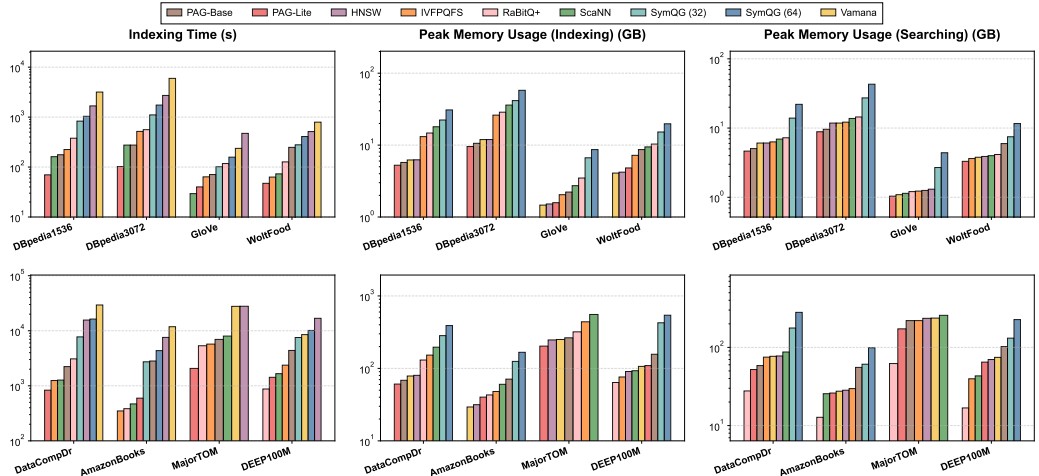

*Figure 4.* Indexing time and peak memory usage. SymQG runs out of memory on MajorTOM.

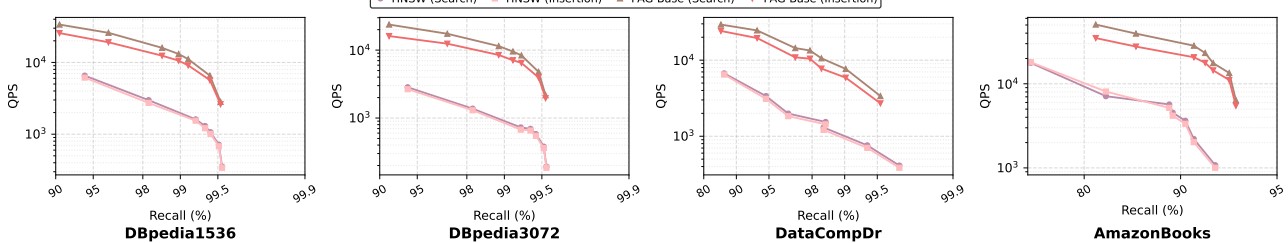

*Figure 5.* QPS-recall of query workloads with online insertions, $K = 100$.

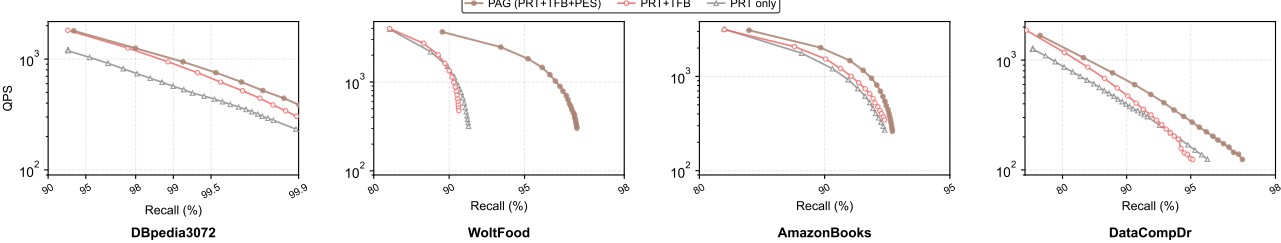

*Figure 6.* Ablation study, QPS-recall. PRT only is the re-implementation of KS2 in PAG.

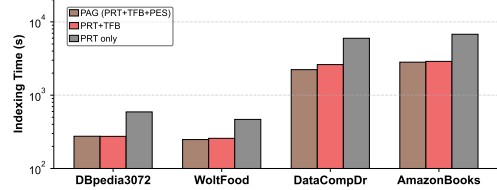

*Figure 7.* Ablation study, indexing time. PRT only is the re-implementation of KS2 in PAG.

We observe that TFB consistently reduces indexing time and improves search performance. PES further enhances search performance with negligible additional indexing time. The additional memory usage introduced by TFB and PES is very minor and thus not shown here, because TFB does not affect the index size and PES barely increases the number of edges.

## 6. Conclusion

In this paper, we motivated six critical demands of modern AI applications for ANNS, covering search performance, indexing speed, memory footprint, scalability to dimensionality, robustness against retrieval size, and support for online insertions. To meet these demands, we proposed PAG as a new framework for ANNS. PAG reduces unnecessary distance computations by employing the comparison of exact and approximate distances. The components of PAG are derived from a unified statistical relationship, making its mechanism theoretically explainable. Experiments on modern datasets showcased the superiority of two PAG variants—one for high QPS and the other for fast indexing and small index size—over widely-used ANNS methods as well as state-of-the-art solutions under the criteria specified by the six demands.

## Acknowledgements

This work is supported by JSPS Kakenhi JP23K17456, JP23K24850, JP23K25157, JP23K28096, JP25H01117, JP25K21272, JP26K02916, JP26K03246, JST CREST JP-MJCR22M2, NSFC 62472289, 62532007, and Guangdong Provincial Key Laboratory of Popular High Performance Computers 2017B030314073.

## Impact Statement

In this paper, we study ANNS for high-dimensional data and propose a new algorithm framework. As the proposed technique is a fundamental algorithmic contribution, it has no direct ethical implications. In the future, we plan to implement the proposed framework in vector databases and expect further research efforts to be built upon this framework.

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

## A. Frequently Used Notations

Table 3 shows the notations frequently used in this paper.

*Table 3.* Frequently used notations.

| Symbol | Description |
|---|---|
| $\mathcal{D}$ | Dataset of vectors |
| $d$ | Dimension of vectors |
| $n$ | Dataset size |
| $\boldsymbol{q}$ | Query vector |
| $K$ | Retrieval size |
| $\boldsymbol{v}$ | Node to be inserted to a graph index |
| $N_{\text{out}}(\boldsymbol{v})$ | Out-neighbors of $\boldsymbol{v}$ |
| $N_{\text{in}}(\boldsymbol{v})$ | In-neighbors of $\boldsymbol{v}$ |
| $\boldsymbol{u}$ | Candidate neighbor of $\boldsymbol{v}$ to be evaluated |
| $L$ | Space partition size |
| $\{\boldsymbol{w_i}\}_{i=1}^t$ | $t$ neighbors of $\boldsymbol{u}$ |
| $\{\boldsymbol{e_i}\}_{i=1}^t$ | Edges between $\boldsymbol{u}$ and $\{\boldsymbol{w_i}\}_{i=1}^t$ |
| $\tau_i$ | Threshold w.r.t. $\boldsymbol{w_i}$ in PRT |
| $\delta_i$ | Threshold w.r.t. $\boldsymbol{w_i}$ in PES |
| $R_L$ | Result/candidate list |
| $W$ | Working set |
| $R_F, R_T$ | Dual rings of false positives and ejected nodes |
| $m$ | Number of projection vectors in each subspace |
| $\mathcal{F}$ | Set of concatenated projection vectors |
| $\{\boldsymbol{r_i}\}_{i=1}^{mL}$ | Concatenated projection vectors in $\mathcal{F}$ |
| $\boldsymbol{r_i^*} \in \mathcal{F}$ | Vector having the smallest angle with $\boldsymbol{w_i} - \boldsymbol{u}$ |
| $\cos \alpha_i$ | Cosine of the angle between $\boldsymbol{w_i} - \boldsymbol{u}$ and $\boldsymbol{v} - \boldsymbol{u}$ |
| $\cos \beta_i$ | Cosine of the angle between $\boldsymbol{w_i} - \boldsymbol{u}$ and $\boldsymbol{r_i^*}$ |
| $\cos \theta_i$ | Cosine of the angle between $\boldsymbol{v} - \boldsymbol{u}$ and $\boldsymbol{r_i^*}$ |
| Cand($\boldsymbol{v}$) | All nodes visited during the insertion of $\boldsymbol{v}$ |
| $H(\boldsymbol{v})$ | Buffer storing PES-surviving candidate edges to $v$ |

## B. Proof of Theorem 3.1

In the proof, with slight abuse of notation, we treat $\boldsymbol{u}$ as the origin, and directly use $\boldsymbol{v}$ and each $\boldsymbol{w_i}$ to denote $\boldsymbol{v} - \boldsymbol{u}$ and $\boldsymbol{w_i} - \boldsymbol{u}$, respectively.

As such, we have $\boldsymbol{w_1}, \ldots, \boldsymbol{w_t}, \boldsymbol{v} \in \mathbb{S}^{d-1}$, and $\langle \boldsymbol{w_i}, \boldsymbol{v} \rangle = \cos \alpha_i, i = 1, \ldots, t$. Suppose that $d$ is divisible by $L$. Let $\boldsymbol{w_i} = [\boldsymbol{w_{i1}}, \ldots, \boldsymbol{w_{iL}}]^\top$ and $\boldsymbol{v} = [\boldsymbol{v_1}, \ldots, \boldsymbol{v_L}]^\top$ be the equal-dimension partition of $\boldsymbol{w_i}$ and $\boldsymbol{v}$, respectively. Suppose that in the $l$-th subspace, we generate $m$ random projection vectors $\{\boldsymbol{r_{jl}}\}_{j=1}^m$ with the norm $1/\sqrt{L}$. We use $\boldsymbol{r_{il}^*}$ to denote the nearest projection vector to $\boldsymbol{w_{il}}$, i.e.,

$$\boldsymbol{r_{il}^*} = \arg\max_{1 \le j \le m} \langle \boldsymbol{r_{jl}}, \boldsymbol{w_{il}} \rangle. \tag{7}$$

Then, we have $\boldsymbol{r_i^*} = [\boldsymbol{r_{i1}^*}, \ldots, \boldsymbol{r_{iL}^*}]^\top \in \mathbb{S}^{d-1}$. We introduce

$$C_{il} = \frac{\langle \boldsymbol{w_{il}}, \boldsymbol{r_{il}^*} \rangle \sqrt{L}}{\|\boldsymbol{w_{il}}\|} \in [-1, 1]. \tag{8}$$

Let $\mathbb{E}[C_{il}] = \mu_m$ and $\text{Var}(C_{il}) = \sigma_m^2$, where $\mu_m$ and $\sigma_m^2$ depend on $m$ and the subspace dimension $d/L$. When $m$ grows at a sufficiently high rate compared to $d/L$, $\mu_m \to 1$ and $\sigma_m \to 0$. Moreover, we have

$$\text{Cov}(C_{il}, C_{jl}) = \rho_m(\phi_{ijl}) \sigma_m^2 \tag{9}$$

where $\phi_{ijl}$ denotes the angle between $\boldsymbol{w_{il}}$ and $\boldsymbol{w_{jl}}$, and $\rho_m : [0, \pi] \to [-1, 1]$ denotes the correlation coefficient function depending on $\phi_{ijl}$. Then, we have

$$Y_i = \sum_{l=1}^L \langle \boldsymbol{r_{il}^*}, \boldsymbol{w_{il}} \rangle = \sum_{l=1}^L \frac{\|\boldsymbol{w_{il}}\|}{\sqrt{L}} C_{il}. \tag{10}$$

By A2, all the $o(1)$ terms below are uniform over $i$ and $l$. Since $\|\boldsymbol{w_{il}}\| = (1 + o(1))/\sqrt{L}$, we have

$$\mathbb{E}[Y_i] = \mu_m(1 + o(1)). \tag{11}$$

$$\text{Var}(Y_i) = \frac{\sigma_m^2}{L}(1 + o(1)). \tag{12}$$

$$\text{Cov}(Y_i, Y_j) = \frac{\sigma_m^2}{L^2} \sum_{l=1}^L \rho_m(\phi_{ijl})(1 + o(1)). \tag{13}$$

Then, $\boldsymbol{v_l}$ can be decomposed as $\boldsymbol{v_l} = \boldsymbol{v_{il}^{\parallel}} + \boldsymbol{v_{il}^{\perp}}$, where $\boldsymbol{v_{il}^{\parallel}}$ and $\boldsymbol{v_{il}^{\perp}}$ are defined as follows.

$$\boldsymbol{v_{il}^{\parallel}} = \frac{\langle \boldsymbol{v_l}, \boldsymbol{w_{il}} \rangle}{\|\boldsymbol{w_{il}}\|^2} \boldsymbol{w_{il}}, \quad \boldsymbol{v_{il}^{\perp}} \perp \boldsymbol{w_{il}}. \tag{14}$$

Then, we can define $Z_i$ as follows.

$$Z_i = \langle \boldsymbol{r_i^*}, \boldsymbol{v} \rangle = Z_i^{(1)} + Z_i^{(2)} \tag{15}$$

where $Z_i^{(1)}$ and $Z_i^{(2)}$ are defined as follows.

$$Z_i^{(1)} = \sum_{l=1}^L \frac{\langle \boldsymbol{v_l}, \boldsymbol{w_{il}} \rangle}{\|\boldsymbol{w_{il}}\| \sqrt{L}} C_{il}. \tag{16}$$

$$Z_i^{(2)} = \sum_{l=1}^L \langle \boldsymbol{r_{il}^*}, \boldsymbol{v_{il}^{\perp}} \rangle. \tag{17}$$

$$\mathbb{E}[Z_i^{(1)}] = \frac{\mu_m}{\sqrt{L}} \sum_{l=1}^L \frac{\langle \boldsymbol{v_l}, \boldsymbol{w_{il}} \rangle}{\|\boldsymbol{w_{il}}\|} \tag{18}$$
$$= \mu_m \cos \alpha_i + o(1).$$

By symmetry, we have $\mathbb{E}[Z_i^{(2)} | \{C_{il}\}_{l=1}^L] = 0$. Then, by the independence of subspaces, we have

$$\mathbb{E}[Z_i] = \mu_m \cos(\alpha_i) + o(1). \tag{19}$$

$$\text{Var}(Z_i^{(1)}) = \frac{\sigma_m^2}{L} \sum_{l=1}^L \frac{\langle \boldsymbol{v_l}, \boldsymbol{w_{il}} \rangle^2}{\|\boldsymbol{w_{il}}\|^2}. \tag{20}$$

We take the orthogonal decomposition of $\boldsymbol{r_{il}^*}$ as $\boldsymbol{r_{il}^*} = \boldsymbol{r_{il}^{\parallel}} + \boldsymbol{r_{il}^{\perp}}$, where $\boldsymbol{r_{il}^{\parallel}}$ has the same direction with $\boldsymbol{w_{il}}$. Let $\boldsymbol{r_{il}^{\perp}} = \|\boldsymbol{r_{il}^{\perp}}\| \boldsymbol{\zeta}$, where $\|\boldsymbol{\zeta}\| = 1$ is a random vector in a $(d/L - 1)$-dimensional subspace. We have

$$\mathbb{E}_{\boldsymbol{\zeta}}[\langle \boldsymbol{\zeta}, \boldsymbol{v_{il}^{\perp}} \rangle^2] = \frac{\|\boldsymbol{v_{il}^{\perp}}\|^2 L}{d - L}. \tag{21}$$

By symmetry, $\mathbb{E}[\langle r_{il}^{\perp}, v_{il}^{\perp}\rangle \mid C_{il}] = 0$. Then, we have

$$\text{Var}(\langle r_{il}^{*}, v_{il}^{\perp}\rangle) = \mathbb{E}_{C_{il}}\left[\mathbb{E}[\langle r_{il}^{\perp}, v_{il}^{\perp}\rangle^2 \mid C_{il}]\right]. \qquad (22)$$

Using $\mathbb{E}[\|r_{il}^{\perp}\|^2] = \frac{1}{L}(1 - \mathbb{E}[C_{il}^2])$, we have

$$\text{Var}(Z_i^{(2)}) = \frac{1 - (\sigma_m^2 + \mu_m^2)}{d - L}\sum_{l=1}^{L}\|v_{il}^{\perp}\|^2. \qquad (23)$$

Because $\text{Cov}(Z_i^{(1)}, Z_i^{(2)}) = 0$, we have

$$\text{Var}(Z_i) = \text{Var}(Z_i^{(1)}) + \text{Var}(Z_i^{(2)}). \qquad (24)$$

We define $\epsilon_m = \sqrt{1 - \mathbb{E}[C_{il}^2]}$. From Eq. (23), we can see that $\text{Var}(Z_i^{(2)}) = O(\epsilon_m^2/L)$.

We now analyze the covariance structure. We decompose the covariance matrix as

$$\text{Cov}(Z_i, Z_j) = \text{Cov}(Z_i^{(1)}, Z_j^{(1)}) + R_{ij} \qquad (25)$$

where the residual term $R_{ij}$ contains the noise auto-covariance and cross-terms:

$$R_{ij} = \text{Cov}(Z_i^{(2)}, Z_j^{(2)}) + \text{Cov}(Z_i^{(1)}, Z_j^{(2)}) + \text{Cov}(Z_i^{(2)}, Z_j^{(1)}). \qquad (26)$$

Because $\text{Var}(Z^{(1)}) = O(1/L)$ and $\text{Var}(Z^{(2)}) = O(\epsilon_m^2/L)$, by the Cauchy-Schwarz inequality, we have

$$|\text{Cov}(Z_i^{(1)}, Z_j^{(2)})| \leq \sqrt{\text{Var}(Z_i^{(1)})\text{Var}(Z_j^{(2)})} = O\left(\frac{\epsilon_m}{L}\right). \qquad (27)$$

Thus, the entire residual term satisfies $R_{ij} = O(\epsilon_m/L)$.

Based on the above analysis, define

$$\mathbf{D} = \text{diag}[\cos\alpha_1, \ldots, \cos\alpha_t]. \qquad (28)$$

For each $i$ and $l$, let

$$Y_{il} = \langle r_{il}^{*}, w_{il}\rangle = \frac{\|w_{il}\|}{\sqrt{L}}C_{il}. \qquad (29)$$

We also define

$$a_{il} = \frac{\langle v_l, w_{il}\rangle}{\|w_{il}\|^2}. \qquad (30)$$

Then, by the definition of $v_{il}^{\parallel}$, we have

$$Z_i = \sum_{l=1}^{L}a_{il}Y_{il} + \sum_{l=1}^{L}\langle r_{il}^{*}, v_{il}^{\perp}\rangle. \qquad (31)$$

Therefore,

$$Z_i - \cos\alpha_i Y_i = \sum_{l=1}^{L}(a_{il} - \cos\alpha_i)Y_{il} + \sum_{l=1}^{L}\langle r_{il}^{*}, v_{il}^{\perp}\rangle. \qquad (32)$$

By A2, uniformly over $i$ and $l$,

$$|a_{il} - \cos\alpha_i| = O(\eta_L). \qquad (33)$$

Since $|C_{il}| \leq 1$ and $\|w_{il}\| = (1 + o(1))/\sqrt{L}$, we have

$$\sum_{l=1}^{L}|Y_{il}| \leq \sum_{l=1}^{L}\frac{\|w_{il}\|}{\sqrt{L}} = 1 + o(1). \qquad (34)$$

Thus, the first term in Eq. (32) satisfies

$$\left|\sum_{l=1}^{L}(a_{il} - \cos\alpha_i)Y_{il}\right| = O(\eta_L). \qquad (35)$$

The second term in Eq. (32) is the orthogonal residual analyzed above. By Eq. (23), we have

$$\text{Var}\left(\sum_{l=1}^{L}\langle r_{il}^{*}, v_{il}^{\perp}\rangle\right) = O(\epsilon_m^2/L). \qquad (36)$$

In addition, since

$$\text{Var}\left(\sum_{l=1}^{L}(a_{il} - \cos\alpha_i)Y_{il}\right) = O(\eta_L^2\sigma_m^2/L), \qquad (37)$$

the Cauchy-Schwarz inequality gives the covariance bound for the residual terms. For fixed $t$, if we define

$$\omega_{m,L} = \eta_L + \epsilon_m, \qquad (38)$$

then the residual covariance satisfies

$$\left\|\text{Cov}\left([Z_1 - \cos\alpha_1 Y_1, \ldots, Z_t - \cos\alpha_t Y_t]^{\top}\right)\right\| = O(\omega_{m,L}/L). \qquad (39)$$

Now define the block residual vector

$$\boldsymbol{\Delta_l} = [\Delta_{1l}, \ldots, \Delta_{tl}]^{\top}, \qquad (40)$$

where

$$\Delta_{il} = (a_{il} - \cos\alpha_i)Y_{il} + \langle r_{il}^{*}, v_{il}^{\perp}\rangle. \qquad (41)$$

Then Eq. (32) can be written in vector form as

$$\mathbf{Z} = \mathbf{D}\mathbf{Y} + \sum_{l=1}^{L}\boldsymbol{\Delta_l}, \qquad (42)$$

where $\mathbf{Z} = [Z_1, \ldots, Z_t]^{\top}$ and $\mathbf{Y} = [Y_1, \ldots, Y_t]^{\top}$.

The above decomposition isolates the leading geometric term $\mathbf{D}\mathbf{Y}$ from the projection statistic. The preceding moment and covariance bounds further show that the remaining residual has a controlled scale and arises from the aggregation of block-level fluctuations over independent subspaces. We now

state the residual regularity assumptions referred to in Theorem 3.1. The projection codebooks are independent across subspaces and rotationally invariant within each subspace. Moreover, for $\mathbf{Y}$ in its typical set, the residual triangular array $\{\boldsymbol{\Delta}_l\}_{l=1}^{L}$ satisfies a uniform conditional multivariate normal approximation:

$$\mathcal{L}\Big(\sum_{l=1}^{L}\big(\boldsymbol{\Delta}_l - \mathbb{E}[\boldsymbol{\Delta}_l \mid \mathbf{Y}]\big)\Big|\mathbf{Y}\Big) = \mathcal{N}(0, \bar{\Sigma}_{m,L}) + o(1),$$

$$(43)$$

where the $o(1)$ term is in bounded-Lipschitz distance and the approximation is uniform over the typical set. The typical set is chosen so that the conditional approximation is uniform and $\|\mathbf{Y} - \boldsymbol{\mu}_Y\| = O(1)$, where $\mu_Y = \mathbb{E}[Y]$. This holds with probability tending to one for fixed $t$. The covariance matrix in the above approximation satisfies

$$\|\bar{\Sigma}_{m,L}\| = O(\omega_{m,L}/L). \qquad (44)$$

Let

$$\boldsymbol{r}_L = \mathbb{E}\left[\sum_{l=1}^{L}\boldsymbol{\Delta}_l \mid \mathbf{Y}\right]. \qquad (45)$$

As part of the residual regularity assumptions, we require that, uniformly over the same typical set,

$$\|\boldsymbol{r}_L\| = O(\eta_L). \qquad (46)$$

The one-block rotational symmetry motivates this mean-regularity condition, but we do not require an exact zero conditional mean after conditioning only on $\mathbf{Y}$.

Combining the residual decomposition with the conditional normal approximation gives

$$\mathcal{L}(\mathbf{Z} \mid \mathbf{Y}) = \mathcal{N}(\mathbf{DY} + \boldsymbol{r}_L, \bar{\Sigma}_{m,L}) + o(1). \qquad (47)$$

Since $\mathbf{Z} = \mathbf{X}$ and $\mathbf{Y} = [\cos\beta_1, \ldots, \cos\beta_t]^{\top}$, we have

$$\mathbf{DY} = [\cos\alpha_1\cos\beta_1, \ldots, \cos\alpha_t\cos\beta_t]^{\top}. \qquad (48)$$

Equivalently, conditioning on $\{\alpha_i, \beta_i\}_{i=1}^{t}$ gives

$$\mathcal{L}\big(\mathbf{X} \mid \{\alpha_i, \beta_i\}_{i=1}^{t}\big) = \mathcal{N}(\boldsymbol{\mu}_{\alpha,\beta} + \boldsymbol{r}_L, \bar{\Sigma}_{m,L}) + o(1), \qquad (49)$$

where

$$\boldsymbol{\mu}_{\alpha,\beta} = [\cos\alpha_1\cos\beta_1, \ldots, \cos\alpha_t\cos\beta_t]^{\top}. \qquad (50)$$

This proves Theorem 3.1.

**Remarks.** For fixed $t$, suppose that a common random rotation matrix is applied to $\boldsymbol{v} - \boldsymbol{u}$ and all $\boldsymbol{w}_i - \boldsymbol{u}$ in $\mathbb{R}^d$. By the spherical concentration inequality, for $\boldsymbol{x} \in \{\boldsymbol{w}_i - \boldsymbol{u}, \boldsymbol{v} - \boldsymbol{u}\}$, we have

$$P\left(\big|L\|\boldsymbol{x}_l\|^2 - 1\big| \geq \tilde{\epsilon}\right) \leq 2e^{-c_1(d/L)\tilde{\epsilon}^2} \qquad (51)$$

$$P\left(\big|L\langle(\boldsymbol{v} - \boldsymbol{u})_l, (\boldsymbol{w}_i - \boldsymbol{u})_l\rangle - \cos\alpha_i\big| \geq \tilde{\epsilon}\right) \leq 2e^{-c_2(d/L)\tilde{\epsilon}^2}$$

$$(52)$$

where $c_1$ and $c_2$ are constants. By a union bound over all $l$ and fixed $t$, if $L \to \infty$ and $(d/L)/\log L \to \infty$, then

$$\max_{i,l}\left|\sqrt{L}\|(\boldsymbol{w}_i - \boldsymbol{u})_l\| - 1\right| = o_p(1),$$

$$\max_{l}\left|\sqrt{L}\|(\boldsymbol{v} - \boldsymbol{u})_l\| - 1\right| = o_p(1) \qquad (53)$$

$$\max_{i,l}\big|L\langle(\boldsymbol{v} - \boldsymbol{u})_l, (\boldsymbol{w}_i - \boldsymbol{u})_l\rangle - \cos\alpha_i\big| = o_p(1). \qquad (54)$$

Therefore, A2 is mild after a random rotation. This suggests choosing $L$ so that both $L$ and $d/L$ are large; $L = \sqrt{d}$ is a balanced choice, which is consistent with the setting in our experiments.

## C. Related Work

We first supplement Sec. 1.1 with more discussions, and then introduce the preliminaries on probabilistic routing.

### C.1. Discussions on ANNS Solvers

#### C.1.1. VECTOR QUANTIZATION

Quantization-based methods for ANNS have a long history. Representative approaches can be found in (Jégou et al., 2011; Ge et al., 2014; Babenko & Lempitsky, 2016; Guo et al., 2020a; Gao & Long, 2024; Gao et al., 2025). Here, we briefly introduce the basic idea of the most widely-used method, Product Quantization (PQ) (Jégou et al., 2011).

Let $\mathcal{D} \subset \mathbb{R}^d$ be a $d$-dimensional dataset and let $\boldsymbol{q} \in \mathbb{R}^d$ be a query vector. PQ divides each data vector $\boldsymbol{x} \in \mathcal{D}$ into $L$ sub-vectors, i.e., $\boldsymbol{x} = (\boldsymbol{x}_1, \boldsymbol{x}_2, \ldots, \boldsymbol{x}_L)$, where each sub-vector $\boldsymbol{x}_j \in \mathbb{R}^{d'}$ has dimension $d' = d/L$. In this way, PQ constructs $L$ subspaces of dimension $d'$, each containing the corresponding sub-vectors of all data points.

In each subspace, every sub-vector is assigned to one of $2^k$ quantized sub-vectors. Equivalently, each sub-vector is represented by a sub-codeword of length $k$ bits. The set of all sub-codewords in the $j$-th subspace is referred to as a codebook and is denoted by $\mathcal{C}_j$. By concatenating the sub-codewords across all subspaces, each original vector is represented by a codeword of length $kL$, taking values in the Cartesian product $\mathcal{C}_1 \times \cdots \times \mathcal{C}_L$.

During the query phase, PQ first computes the distances between the query $\boldsymbol{q}$ and all quantized sub-vectors in each subspace. For each data vector $\boldsymbol{x}$, PQ then sums the corresponding $L$ subspace distances to obtain an approximate distance between $\boldsymbol{x}$ and $\boldsymbol{q}$. After computing the approximate distances for all data vectors, PQ ranks them accordingly and returns the most promising candidates.

### C.1.2. SIMILARITY GRAPH

The workflow of graph-based methods is roughly as follows. In the indexing phase, a similarity graph is constructed as the index structure, where data points serve as nodes and edges connect pairs of nearby nodes. A search can move directly from one node to another only if an edge exists between them. In the query phase, the node with the highest priority in a priority queue is selected, and all of its connected neighbors are visited. During this process, the priority queue is updated whenever a node closer to the query is discovered. The search ends when all nodes in the priority queue have been visited. Finally, the top-$K$ points in the priority queue are returned.

Among various graph-based methods, HNSW (Malkov & Yashunin, 2020) is a notable one that employs a multi-layer hierarchical structure to achieve rapid routing. NSG (Fu et al., 2019) optimizes the graph topology to ensure the existence of monotonic paths towards a central entry point, thereby enhancing search efficiency. Vamana, which was introduced along with DiskANN (Subramanya et al., 2019), iteratively refines a random graph into a high-performance graph structure. Despite their structural differences, these methods converge on a similar edge selection criterion, namely **RobustPrune**, which prioritizes directional diversity over simple proximity to ensure efficient navigation through high-dimensional space.

In addition to the three graphs mentioned above, many graph-based methods have been proposed recently (Lu et al., 2021; Gao & Long, 2023; Xie et al., 2025; Wang et al., 2025b). Despite different designs, most of them rely on existing graphs, such as HNSW.

### C.1.3. QUANTIZED GRAPH

As stated in the main body of this paper, QG can achieve high performance on some datasets for small values of $K$, and in certain cases can be $4\times - 10\times$ faster than HNSW. However, such improvement is often not robust and suffers from several limitations: (1) **Sensitivity to data distribution.** The effectiveness of QG relies on the assumption that the ranking induced by quantized distances is sufficiently close to the ranking under the original distances. Unfortunately, this assumption often does not hold, especially for many modern real-world datasets where semantic embeddings are increasingly complex. (2) **Sensitivity to $K$.** As $K$ increases, the performance of QG degrades significantly. When $K$ reaches the thousand scale, QG generally has no significant advantage over HNSW. (3) **Very large space cost.** To improve the accuracy of QG, more bits are required to represent the quantized vectors. As a result, the memory consumption of QG is typically at least $2\times$ larger than HNSW, compromising the use of QG for large datasets.

## C.2. Preliminaries of Routing Test

### C.2.1. RANDOM PROJECTION FOR ANGLE ESTIMATION

Because the norms can be pre-computed, estimating the $\ell_2$ distance is equivalent to estimating the cosine of the angle between vectors. In high-dimensional Euclidean spaces, angle estimation via random-projection techniques has been extensively studied, with Locality Sensitive Hashing (LSH) (Indyk & Motwani, 1998; Andoni & Indyk, 2005; 2008) being one of the most influential approaches. Among various LSH methods, SimHash (Charikar, 2002) is a representative one. Its core idea is to generate multiple random hyperplanes that partition the space into cells, so that vectors falling into the same cell are likely to form a small angle with each other. Subsequent studies have proposed more refined strategies for angular distance estimation. In particular, Andoni et al. (2015) introduced Falconn, an LSH method that identifies the projection vector yielding the largest or smallest projection value for a given data vector and uses the corresponding projection index as the hash value. This design leads to substantially improved search performance compared to SimHash. Building on this idea, Pham (2021) further incorporated Concomitants of Extreme Order Statistics (CEOs) to explicitly identify the projection achieving the maximum or minimum inner product with the data vector, and to record the associated extreme projected value. By exploiting this additional information beyond a discrete hash value, a more accurate estimation of angular distance can be achieved (Pham & Liu, 2022).

### C.2.2. ROUTING TEST IN SIMILARITY GRAPHS

Due to its simplicity and ease of implementation, CEOs has been adopted in a variety of similarity search tasks (Pham, 2021; Andoni et al., 2015; Xu & Pham, 2024). Beyond standalone similarity estimation, CEOs has also been leveraged to accelerate similarity graphs, which constitute one of the most effective structures for ANNS. By swapping the roles of the query and data vectors in the original CEOs formulation, Lu et al. (2024) developed a space-partitioning technique and proposed the PEOs test. This test enables probabilistic comparisons between the objective angle and a fixed threshold, and has been integrated into the routing procedures of similarity graphs. Specifically, for every visited node $u$, we send all the out-neighbors of $u$ to PEOs. Only the exact distances between $q$ and the nodes that pass the PEOs test are computed. In the experiments of (Lu et al., 2024), only around 25% nodes can pass the PEOs test. As a result, substantial improvements in search performance brought by PEOs were reported over similarity graphs such as HNSW (Malkov & Yashunin, 2020) and NSSG (Fu et al., 2022). More recently, Lu et al. (2025) proposed a new test function, KS2, which achieves higher test accuracy while maintaining a shorter test time compared with PEOs. KS2 employs a projection

structure similar to that of PEOs, but additionally incorporates the reference angle into the testing procedure. Lu et al. (2025) further showed that, without introducing additional assumptions, the test guarantees a success probability of at least $0.5$ when deciding whether the exact distance between the objective node and $q$ should be evaluated.

## D. Pseudo-codes of PAG

The pseudo-codes of the PAG algorithms are presented in Algs. 1 and 2, which represent the indexing and searching, respectively. Note that indexing is essentially the sequential insertion of all nodes. Similar to (Lu et al., 2024), we store the computed projections in a projection table for lookup. For each stored edge $(u, w_i)$, the inference structure $I$ stores the selected block-level reference indices, $\cos \beta_i$, and the offset projections $\{\langle u_l, r_{il}^* \rangle\}_{l=1}^{L}$. Thus, the projection table built for $v$ or $q$ is converted to the relative projection required by PRT and PES by subtracting these stored offsets.

## E. Parameter Analysis of PAG

We analyze the settings of the parameters in PAG.

(1) $m$: The value of $m$ is fixed to 16, so that the selected projection index in each subspace can be represented by 4 bits, which is compatible with the AVX512 instruction set. The choice of 4 bits follows NGT-QG (Yahoo! Japan, 2023).

(2) $L$: $L$ is recommended to be in $[8, d/8]$, where 8 is compatible with AVX512 because at most 8 levels can be accessed at a time, while $d/L = 8$ is a safe ratio that ensures the accuracy of the multi-level projection structure, as explained in PEOs (Lu et al., 2024). Based on the analysis in Sec. 4.2, $L$ is used to tune the trade-off between efficiency and accuracy.

(3) $M$: Similar to its usage in HNSW, $2M$ is the maximum out-degree of PAG. Based on the analysis in Sec. 4.2, $M$ is recommended to be 64 when space cost permits. In practice, $M$ is generally chosen from $\{16, 32, 64\}$.

(4) $|W|$: We denote the size of $W$ by $b$. During indexing, we set $b = \min\{efC, 100\}$ in all experiments. During search, we set $b = \max\{10, K\}$, so that when $K \geq 10$, $W$ is aligned with the result list and refilling can be easily executed.

(5) $efC$: $efC$ determines how many nodes are visited in $W$ and plays a role similar to $efC$ in HNSW. It is used to control the trade-off between indexing time and search performance. For fast indexing, $efC$ is recommended to be in $[100, 200]$. For high graph quality, $efC$ is recommended to be in $[1000, 10000]$. Notably, thanks to TFB, PAG with a very large $efC$ can be built much faster than HNSW with the same $efC$.

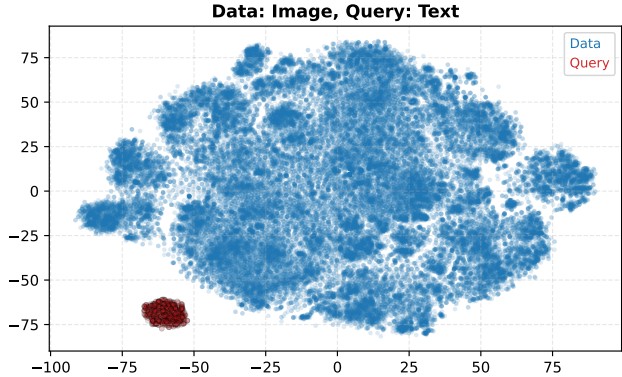

*Figure 8.* DataCompDr data (sampled 50,000 vectors) and query (1,000 vectors) distributions.

## F. Experimental Setup Details

### F.1. Environment

All experiments were conducted on a machine equipped with an Intel Xeon Platinum 8276L CPU, which supports AVX-512 instructions and provides 112 hardware threads. The system was configured with 754 GB of DDR4 ECC memory and runs Ubuntu 22.04. For indexing, all 112 threads were used, while search was performed using a single CPU thread, in line with the standard setting in ANN-Benchmarks (Bernhardsson et al., 2015).

### F.2. Baselines

F.2.1. SELECTION OF BASELINES

For quantization-based methods, we choose IVF-PQFS (Jégou et al., 2011), ScaNN (Guo et al., 2020a), and RaBitQ+ (Gao et al., 2025) as baselines, where RaBitQ+ (Gao et al., 2025) is an improved version of RaBitQ (Gao & Long, 2024). For graph-based methods, we choose HNSW (Malkov & Yashunin, 2020), Vamana (Subramanya et al., 2019), and SymphonyQG (Gou et al., 2025a), where SymphonyQG has shown its superiority over LVQ (Aguerrebere et al., 2023) and NGT-QG (Yahoo! Japan, 2023). On the other hand, HNSW+KS2 (Lu et al., 2025) has been shown to perform better than HNSW+PEOs (Lu et al., 2024), and KS2 can be approximately viewed as our PRT without TFB, modulo threshold-setting differences. Thus, for a fair comparison, we re-implement the KS2 component within our framework and report results for PAG versus PRT only in our ablation study. For other baselines, we use existing implementations: IVFPQFS (Douze et al., 2026), ScaNN (Guo et al., 2025), RaBitQ+ (Gao et al., 2026), HNSW (Malkov & Yashunin, 2026), Vamana (Simhadri et al., 2026), and SymphonyQG (Gou et al., 2025b).

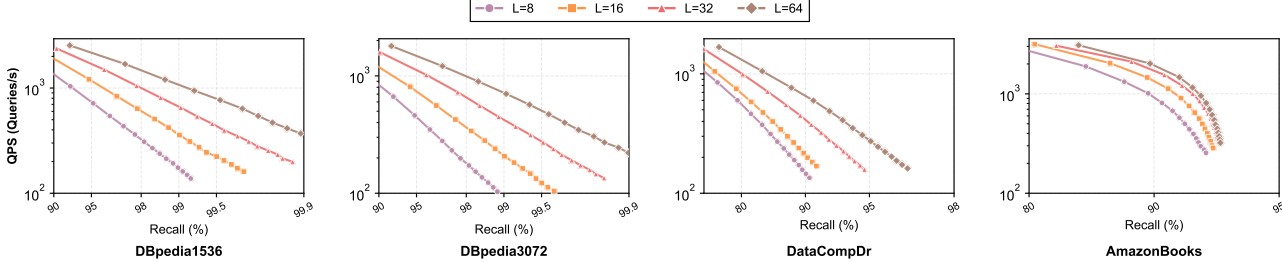

*Figure 9.* QPS-recall of PAG-Base under varying space partition size $L$, $K = 100$. $M$ and $efC$ values are given in Table 4, PAG-Base.

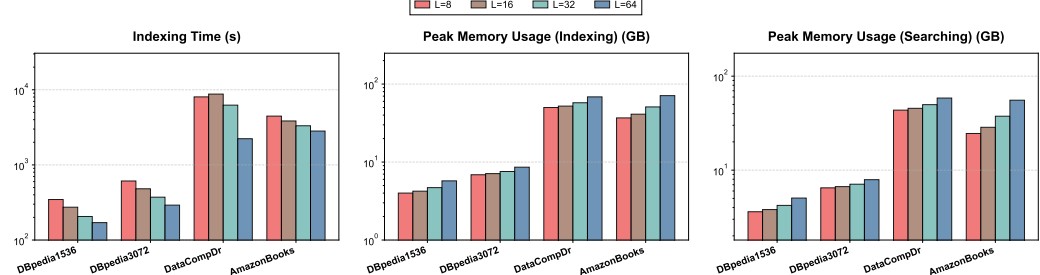

*Figure 10.* Indexing time and peak memory usage of PAG-Base under varying space partition size $L$. $M$ and $efC$ values are given in Table 4, PAG-Base.

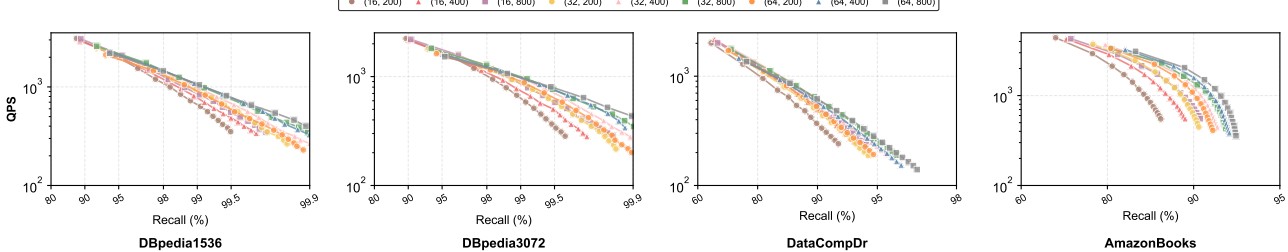

*Figure 11.* QPS-recall of PAG-Base under varying $(M, efC)$, $K = 100$. $L$ values are given in Table 4, PAG-Base.

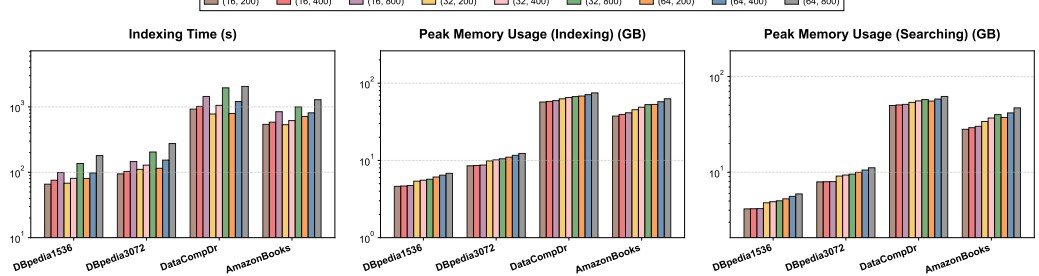

*Figure 12.* Indexing time and peak memory usage of PAG-Base under varying $(M, efC)$. $L$ values are given in Table 4, PAG-Base.

### F.2.2. PARAMETER SETTING

For graph-based methods, we used large graph construction parameters to ensure their best search performance. To ensure a fair comparison of indexing time, the $efC$ parameter in PAG-Base was set to a similar or even larger value than HNSW.

(1) **HNSW**: $efC = 1024$. $M = 32$. The only exception is MajorTOM, where $efC = 512$ due to the long indexing time on this dataset.

(2) **Vamana**: For Vamana, we denote its search-list parameter by $L_{\text{vam}}$. By default, $R_{\text{vam}} = 64$, $L_{\text{vam}} = 1024$, and $\alpha = 1.2$. For better QPS-recall performance on MajorTOM and DEEP100M, we choose $R_{\text{vam}} = 100$, $L_{\text{vam}} = 250$ on MajorTOM, and $R_{\text{vam}} = 110$, $L_{\text{vam}} = 200$ on DEEP100M.

(3) **SymQG**: $efC = 1024$. There are two options for the degree, 32 and 64, resulting in two baselines, SymQG (32) and SymQG (64).

(4) **ScaNN**: We adopt the recommended settings from its

GitHub repository (Guo et al., 2020b).

(5) **RaBitQ+**: Following the same setting as in (Gao et al., 2025), $b = 8$, $k = 16384$ for DataCompDr and 4096 for the other datasets.

(6) **IVFPQFS**: We test combinations of $nlist \in \{1024, 4096, 16384\}$, $k\_factor \in \{64, 128, 256\}$, and $M\_candidates \in [48, 384]$. A combination is chosen for good QPS-recall performance on each dataset.

(7) **PAG**: PAG has two versions, PAG-Base and PAG-Lite. Their parameter settings are listed in Table 4. Users can adjust the values of $efC$, $M$, and $L$ by taking into account the indexing time, memory footprint, and search efficiency.

## G. Additional Experimental Results

### G.1. Effect of Space Partition Size $L$

From the QPS-recall results in Fig. 9, we can see that, within a moderate range, increasing $L$ leads to better search performance, at the cost of a higher memory footprint. As shown in Fig. 10 (left), larger $L$ values also result in longer indexing time, because each projection-based test scans more subspaces and the inference structure becomes larger. On the other hand, this causes a moderately larger index size, which is reflected in the memory footprint reported in Fig. 10 (middle and right). As analyzed in Appendix B, $L = \sqrt{d}$ is an appropriate choice.

### G.2. Effect of $M$ and $efC$

We vary parameters $M$ and $efC$ and plot the QPS-recall in Fig. 11. Larger values of $M$ and $efC$ result in faster query processing speed, and the advantage is more remarkable when users require high recall values. As shown in Fig. 12, increasing $M$ and $efC$ generally leads to more indexing time as well as larger memory footprint. This is expected, because they control the out-degree and the number of nodes visited for each insertion. The only exception is DataCompDr, where a larger $M$ does not necessarily mean a slower indexing speed. This is because the search speed for each inserted vector is accelerated despite a larger out-degree. In general, we suggest using $efC \geq 800$ for fast search and $efC \leq 200$ for fast indexing and small memory footprint, while $efC = 400$ strikes a balance. $M$ is suggested to be 32 or 64.

### G.3. Evaluation on Additional Legacy Datasets

We report the results on four additional legacy datasets, Word2Vec, ImageNet, GIST, and SIFT10M, which are commonly used for ANNS evaluation. Figs. 16, 17, and 18 show the QPS-recall performances when $K = 10, 100$, and 1000. Fig. 19 shows the indexing time and memory footprint. From the results shown in these figures, we have the following observations. PAG-Base maintains its competitiveness in QPS-recall performance, as we have witnessed in the experiments on other datasets. On Word2Vec and ImageNet, where graph-based methods and SymQG struggle to achieve high recalls, PAG-Base has a significant advantage in QPS. On GIST and SIFT10M, SymQG performs better than PAG-Base in QPS when $K = 10$ and $K = 100$, because the two datasets are sparse and well-suited for vector quantization. When $K = 1000$, PAG-Base outperforms SymQG by a large margin. For indexing speed, PAG-Base is slower than graph-based methods and SymQG on Word2Vec but is faster on the other three datasets. PAG-Base also uses less memory than SymQG. The QPS-recall performance of PAG-Lite is generally better than HNSW and Vamana. Meanwhile, PAG-Lite achieves the smallest indexing time, and is competitive in memory footprint.

### G.4. Peak Memory Breakdown

Figs. 20, 21, 22, and 23 report the memory footprint breakdown during indexing and searching. We decompose the peak memory consumption into dataset payload, index structure, and additional overhead. These results provide a more fine-grained view of memory usage than total memory alone. The dataset payloads of PAG methods are smaller because we use 16-bit floats to store the original vectors.

### G.5. Further Comparison with HNSW and Vamana

We compare PAG-Base with HNSW and Vamana under the same sets of graph construction parameter ($efC$ for PAG-Base and HNSW, $L_{\text{vam}}$ for Vamana) settings. $\{200, 400, 800\}$ are tested for these methods. As shown in Fig. 24, when using the same $efC$ or $L_{\text{vam}}$ setting, PAG-Base consistently achieves better QPS-recall performance than HNSW and Vamana, showing that its advantage does not rely on a more favorable graph construction setting. Fig. 25 shows that PAG-Base builds the graph index faster under the same parameter values. Figs. 26 and 27 report the corresponding memory footprint breakdown. We use 16-bit floats to store the original vectors for PAG methods, resulting in smaller dataset payloads than HNSW and Vamana.

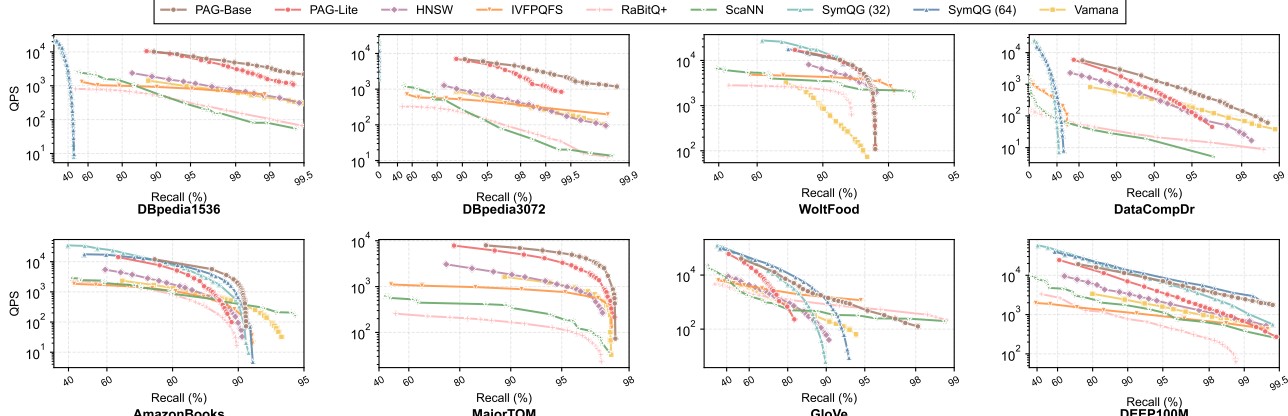

*Figure 13.* QPS-recall, $K = 10$.

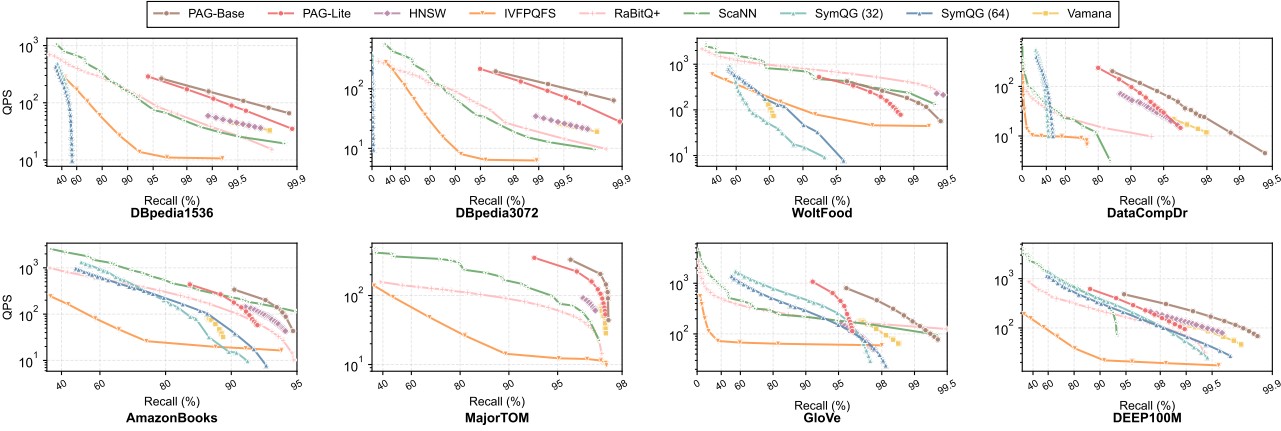

*Figure 14.* QPS-recall, $K = 1000$.

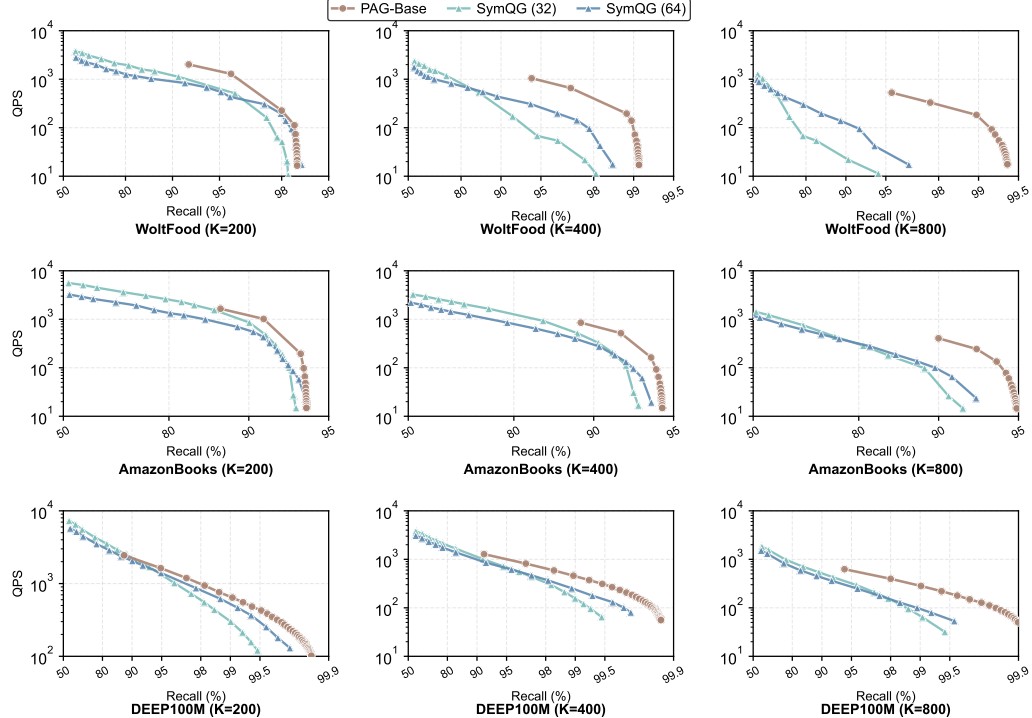

*Figure 15.* QPS-recall comparison of PAG-Base and SymQG under varying retrieval size $K$.

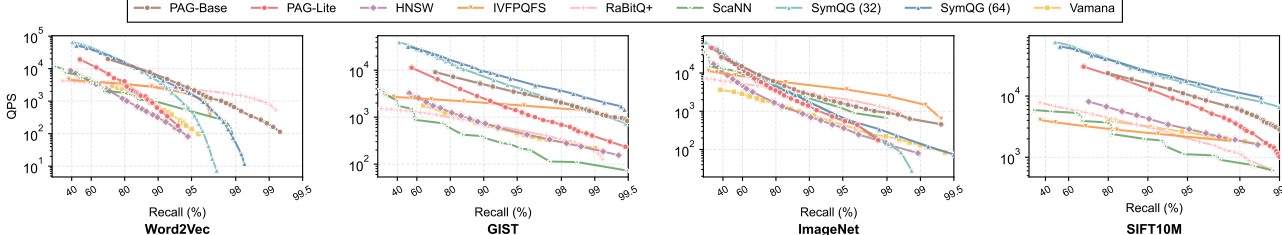

*Figure 16.* QPS-recall on additional legacy datasets, $K = 10$.

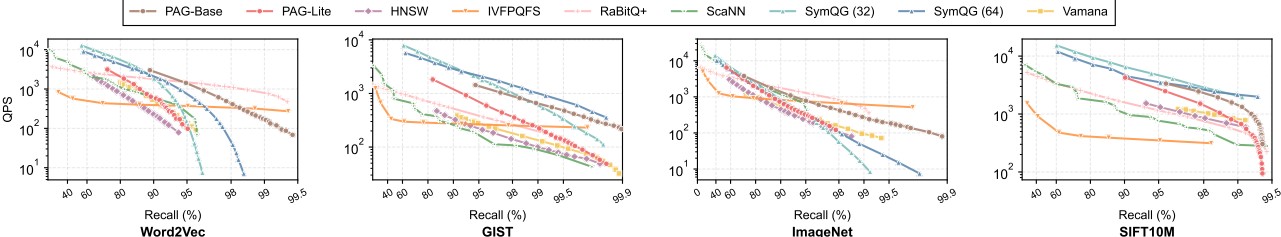

*Figure 17.* QPS-recall on additional legacy datasets, $K = 100$.

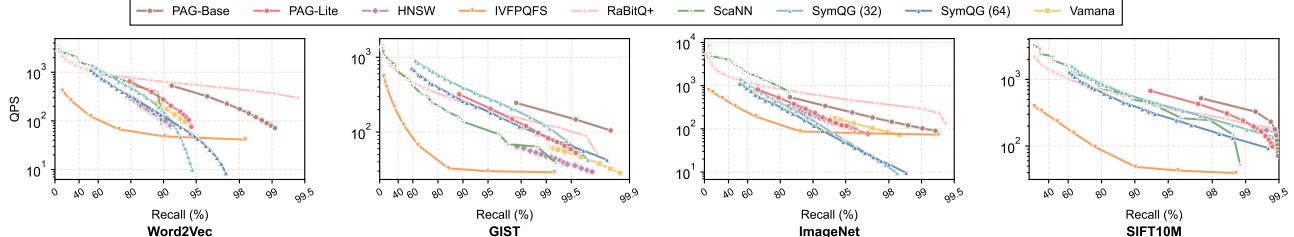

*Figure 18.* QPS-recall on additional legacy datasets, $K = 1000$.

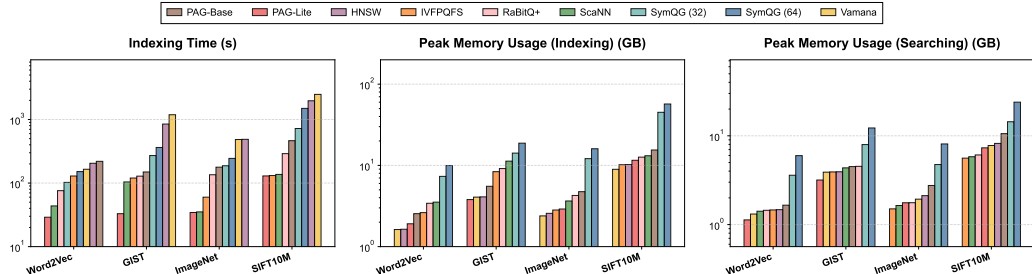

*Figure 19.* Indexing time and peak memory usage on additional legacy datasets.

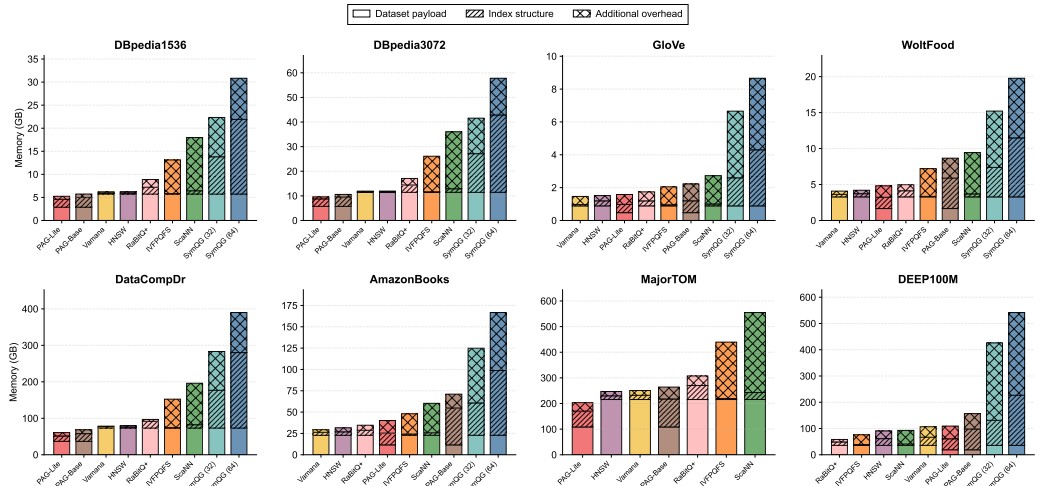

*Figure 20.* Peak memory usage breakdown in the indexing phase, modern datasets.

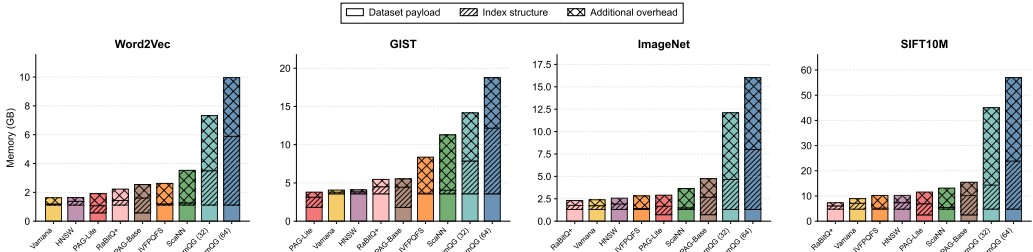

*Figure 21.* Peak memory usage breakdown in the indexing phase, legacy datasets.

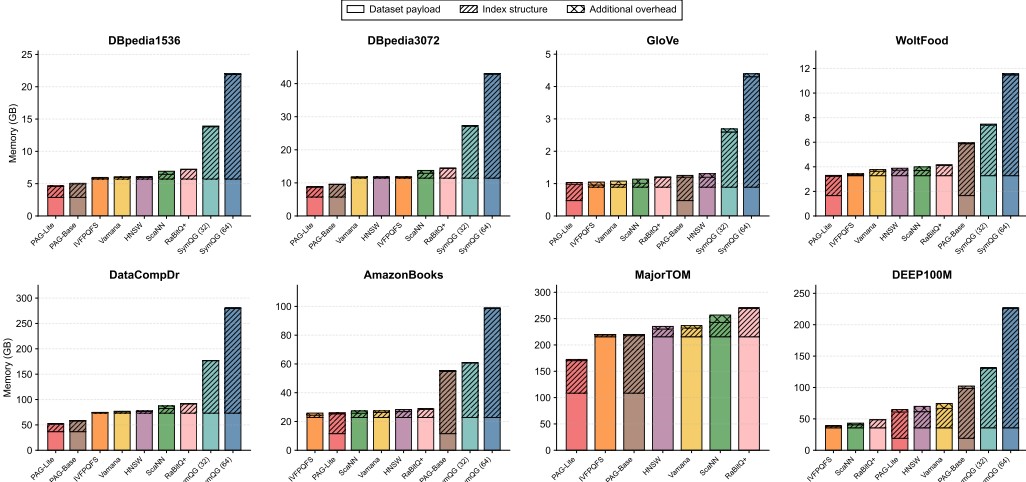

*Figure 22.* Peak memory usage breakdown in the searching phase, modern datasets.

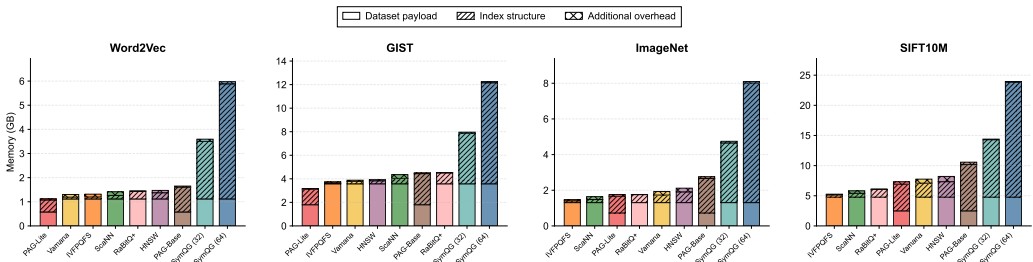

*Figure 23.* Peak memory usage breakdown in the searching phase, legacy datasets.

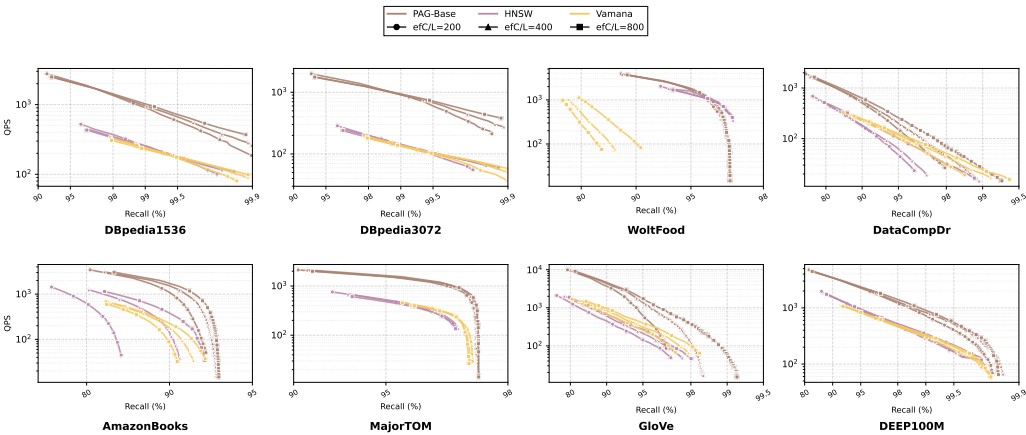

*Figure 24.* QPS-recall comparison of PAG-Base, HNSW, and Vamana under the same sets of $efC$ / $L_{\mathrm{vam}}$.

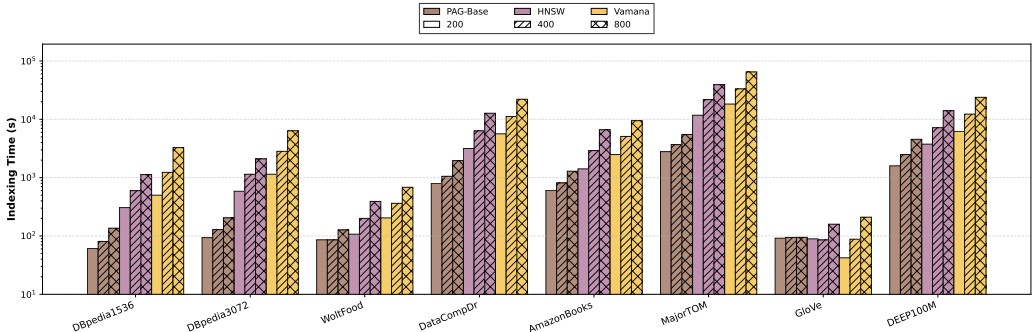

*Figure 25.* Indexing time comparison of PAG-Base, HNSW, and Vamana under the same sets of $efC$ / $L_{\text{vam}}$.

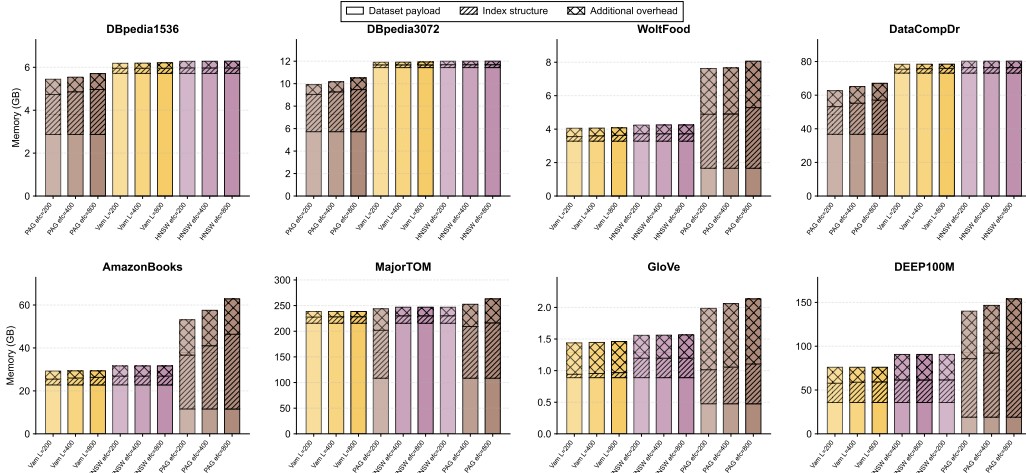

*Figure 26.* Peak memory usage breakdown of PAG-Base, HNSW, and Vamana under the same sets of $efC$ / $L_{\text{vam}}$, indexing phase.

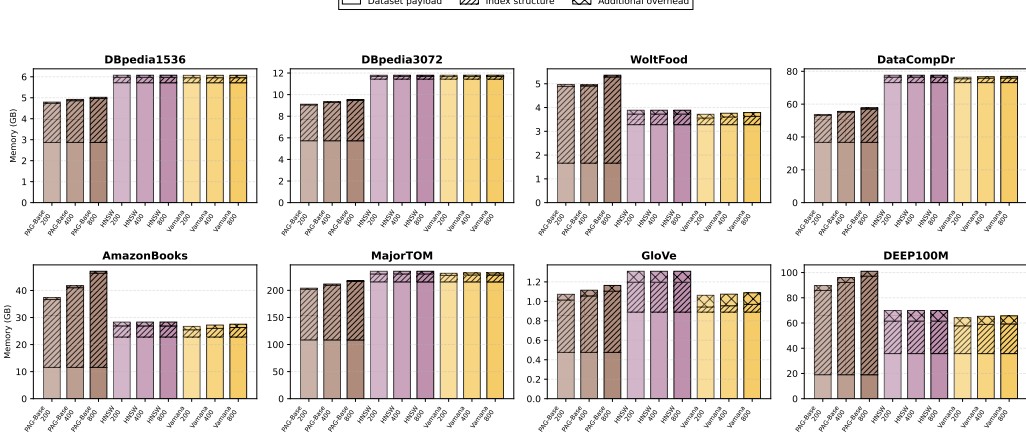

*Figure 27.* Peak memory usage breakdown of PAG-Base, HNSW, and Vamana under the same sets of $efC$ / $L_{\text{vam}}$, searching phase.

**Algorithm 1:** Construction of PAG

**Input:** $G = \emptyset$ is the graph to be built; $\mathcal{D}$ is the dataset; $2M$ is the maximum out-degree; $L$ is the space partition size; $m$ is the number of projection vectors in each subspace; $efC$ is the maximum number of visited nodes; $b$ is the size of $W$

**Output:** $G$ with node set $\mathcal{D}$, edge set $E$, inference structure $I$, transform $Q$, and codebooks $\{\mathcal{C}_l\}_{l=1}^{L}$

1 Initialize $H(\boldsymbol{x}) \leftarrow \emptyset$ for all $\boldsymbol{x} \in \mathcal{D}$, where each $H(\boldsymbol{x})$ has capacity $2M$;

2 The capacity of $R_L$ is set to $efC$;

3 Maximum round $R_{\max}$ is set to $\lceil efC/b \rceil$;

4 Generate a fixed random orthogonal transform $Q$ and replace every $\boldsymbol{x} \in \mathcal{D}$ by $Q\boldsymbol{x}$;

5 Generate block-level codebooks $\{\mathcal{C}_l\}_{l=1}^{L}$, where $\mathcal{C}_l = \{\boldsymbol{r}_{jl}\}_{j=1}^{m}$;

6 Select $b$ initial nodes, connect them, and write the projection information w.r.t. the connected edges to $I$. Let $\mathcal{V}$ denote the set of all initial nodes;

7 **foreach** $v \in \mathcal{D} \setminus \mathcal{V}$ **do**

8     Initialize $W$, $R_F$, $R_T$, and $R_L$ as empty containers;

9     Insert $b$ entry nodes into $W$ and compute their distances to $v$;

10     Compute all $\langle v_l, \boldsymbol{r}_{jl} \rangle$'s and build a projection table;

11     **for** $r = 1$ **to** $R_{\max}$ **do**

12        **if** $W = \emptyset$ **then**

13           **break**;

14        **foreach** unvisited $u \in W$ **do**

15           **foreach** $w \in N_{\text{out}}(u)$ **do**

16              **if** $w$ passes the PRT-TFB test **then**

17                 Compute the distance between $v$ and $w$;

18                 **if** $\|w - v\| < \|z_{\max} - v\|$ **then**

19                    Update $W$ and insert $z_{\max}$ into $R_T$;

20              **else**

21                 Insert $w$ into $R_F$;

22           **if** $(u, v)$ survives the PRT-PES screening **then**

23              Add $u$ to $H(v)$;

24              **if** $|H(v)| > 2M$ **then**

25                 Keep the closest $2M$ nodes to $v$ in $H(v)$;

26     Insert the nodes in $W$ into $R_L$, and empty $W$;

27     Sort and merge $R_F$ and $R_T$, and empty $R_F$ and $R_T$;

28     Refill $W$ until it is full using the sorted elements, and refill $R_T$ with the remaining sorted nodes;

29     Apply RobustPrune to $R_L$ to compute $N_{\text{out}}(v)$ and the initial $N_{\text{in}}(v)$;

30     Add the projection information of new edges to $I$;

31 **foreach** $v \in \mathcal{D}$ with $H(v) \neq \emptyset$ **do**

32     Verify the candidates in $H(v)$ by RobustPrune and add accepted edges only to available out-degree slots; existing edges are kept unchanged. Add the projection information of new edges to $I$;

---

**Algorithm 2:** Search with PAG

**Input:** $G(\mathcal{D}, E, I)$ is the constructed similarity graph; $Q$ is the fixed random orthogonal transform; $\{\mathcal{C}_l\}_{l=1}^{L}$ is the random-projection structure; $efS$ is the maximum number of visited nodes; $q$ is the query; $K$ is the retrieval size

**Output:** Top-$K$ ANN nodes

1 The capacity of $R_L$ is set to $K$;

2 $b = \max\{10, K\}$;

3 Maximum round $R_{\max}$ is set to $\lceil efS/b \rceil$;

4 Initialize $W$, $R_F$, $R_T$, and $R_L$ as empty containers;

5 Set $q \leftarrow Qq$;

6 Compute all $\langle q_l, \boldsymbol{r}_{jl} \rangle$'s and build a projection table;

7 Insert $b$ entry nodes into $W$ and compute their distances to $q$;

8 **for** $r = 1$ **to** $R_{\max}$ **do**

9     **if** $W = \emptyset$ **then**

10        **break**;

11     **foreach** unvisited $u \in W$ **do**

12        **foreach** $w \in N_{\text{out}}(u)$ **do**

13           **if** $w$ passes the PRT-TFB test **then**

14              Compute the distance between $q$ and $w$;

15              **if** $\|w - q\| < \|z_{\max} - q\|$ **then**

16                 Update $W$ and insert $z_{\max}$ into $R_T$;

17           **else**

18              Insert $w$ into $R_F$;

19     Insert the nodes in $W$ into $R_L$, and empty $W$;

20     Sort and merge $R_F$ and $R_T$, and empty $R_F$ and $R_T$;

21     Refill $W$ until it is full using the sorted elements, and refill $R_T$ with the remaining sorted nodes;

22 Return the top-$K$ nodes in $R_L$;

---

*Table 4.* PAG parameter settings.

| Dataset | PAG-Base $(efC, M, L)$ | PAG-Lite $(efC, M, L)$ |
|---|---|---|
| **Modern Datasets** | | |
| DBpedia1536 | $(1000, 32, 128)$ | $(100, 32, 128)$ |
| DBpedia3072 | $(1000, 32, 64)$ | $(100, 32, 64)$ |
| WoltFood | $(2000, 128, 64)$ | $(100, 64, 32)$ |
| DataCompDr | $(1000, 32, 64)$ | $(100, 32, 64)$ |
| AmazonBooks | $(2000, 64, 64)$ | $(100, 64, 32)$ |
| MajorTOM | $(1000, 32, 64)$ | $(100, 16, 64)$ |
| **Legacy Datasets** | | |
| Word2Vec | $(8000, 64, 32)$ | $(200, 64, 32)$ |
| GIST | $(1000, 64, 96)$ | $(64, 32, 96)$ |
| GloVe | $(2000, 64, 32)$ | $(100, 32, 32)$ |
| ImageNet | $(2000, 64, 16)$ | $(200, 16, 16)$ |
| SIFT10M | $(1000, 32, 16)$ | $(100, 16, 16)$ |
| DEEP100M | $(1000, 32, 16)$ | $(100, 16, 16)$ |

