# OpenReview forum: "Approximate Nearest Neighbor Search for Modern AI: A Projection-Augmented Graph Approach"
_ICML.cc/2026/Conference — ICML 2026 regular_

### Official Review · Reviewer_7gFw · 2026-03-12

**Soundness:** 3
**Presentation:** 3
**Significance:** 3
**Originality:** 3
**Overall Recommendation:** 4
**Confidence:** 4

**Summary:**

This paper studies approximate nearest neighbor search for modern AI workloads and argues that ANN methods should be evaluated not only by the usual QPS-recall tradeoff at small retrieval sizes, but also by indexing speed, memory footprint, scalability to high dimensionality, robustness across different values of K, and support for online insertions. To address these goals, the paper proposes Projection-Augmented Graph (PAG), a graph-based ANN framework that integrates projection-based probabilistic tests into graph construction and search. The method combines three main components, Probabilistic Routing Test (PRT), Test Feedback Buffer (TFB), and Probabilistic Edge Selection (PES), and is supported by both theoretical analysis and empirical evaluation. Experiments on a broad set of modern datasets suggest that PAG achieves strong search performance while also offering favorable indexing speed and moderate memory usage, with further results on larger retrieval sizes and online insertion workloads.

**Compliance With Llm Reviewing Policy:**

Affirmed.

**Final Justification:**

The authors have addressed my concerns. I am keeping the positive score.

**Key Questions For Authors:**

see D1-D5

**Limitations:**

yes

**Strengths And Weaknesses:**

S1. The paper addresses an important and timely problem. The motivation that modern embedding retrieval workloads require evaluation beyond standard QPS-recall at small K is convincing, and the six-demand framing is practically meaningful.
S2. The method is technically coherent and reasonably substantial. PAG is not just a small heuristic modification, but a structured framework built around PRT, TFB, and PES, with a nontrivial theoretical basis and clear systems motivation.
S3. The empirical evaluation is broad. The paper considers modern high-dimensional datasets, multiple retrieval sizes, indexing time, memory footprint, and online insertion, which makes the evaluation stronger than a narrow search-only benchmark.
D1. The main concern is comparison fairness. The strongest claims depend on tuned systems comparisons, but the paper does not fully establish equal-memory or equal-indexing-budget comparisons across methods, and tuning effort appears somewhat asymmetric across baselines.
D2. The paper positions itself strongly against prior projection+graph approaches, but the closest comparison line is not as directly represented in the main experiments as one would expect. In particular, it is not fully clear from the main results how much of the gain comes from the core projection-augmented idea beyond prior routing-test methods.
D3. Some of the reported efficiency gains may depend materially on the implementation environment, including AVX-512 support and a high-end CPU setup. The paper would benefit from a clearer discussion of portability and hardware sensitivity.
D4. The current implementation is built on top of HNSW graph indexing, and it is unclear how well the proposed design transfers to other graph-based ANN structures such as Vamana, NSG, or NGT. Without experiments beyond the HNSW, the generality of the method is not yet fully established.
D5. Whether PAG is compatible with vector quantization such as RaBitQ, and whether the main theoretical arguments still hold in quantized vectors.

---

> ### Author Rebuttal · Authors · 2026-03-31
>
> We appreciate the reviewer for acknowledging our contributions and giving constructive comments.
>
> > (D1) The main concern is comparison fairness. The strongest claims depend on tuned systems comparisons, but the paper does not fully establish equal-memory or equal-indexing-budget comparisons across methods, and tuning effort appears somewhat asymmetric across baselines.
>
> We respond to this comment in two aspects.
>
> (1) We considered two settings for PAG: PAG-Base, which is designed for high throughput and is compared with other graph-based methods; and PAG-Lite, which is mainly used to demonstrate the adaptability of PAG to scenarios with limited indexing time and small memory consumption, and is therefore mainly compared with quantization-based baselines.
>
> For all baselines, we tuned their parameters so that each baseline could achieve **near-optimal QPS performance** under the current setting, since throughput is the most important criterion. For PAG-Base, we ensured that its graph construction parameter $efC$ was always set to the same or a higher level than that of other baselines, in order to avoid any advantage in indexing time or memory consumption simply originating from the use of a smaller $efC$. The results show that, in most scenarios, PAG achieves either higher throughput or smaller indexing time and memory usage than the other methods, which is sufficient to support our claim.
>
> (2) We did not compare equal-memory or equal-indexing-budget settings for two reasons: First, different methods often operate at substantially different scales in memory consumption and indexing time. Second, memory consumption and indexing time are essentially outputs rather than tunable parameters. It is difficult to prescribe a target memory budget or indexing time and then compare all methods under that constraint, because trying many $efC$ settings for all these methods is very time-consuming.
>
> > (D3) Some of the reported efficiency gains may depend materially on the implementation environment, including AVX-512 support and a high-end CPU setup. The paper would benefit from a clearer discussion of portability and hardware sensitivity.
>
> In recent years, AVX-512 has become a standard optimization technique in advanced ANNS implementations. For HNSW, ScaNN, and SymQG, their official implementations all employ AVX-512 to accelerate search. Therefore, for fair comparison, we evaluated PAG and all baselines in the same CPU environment with AVX-512 support.
>
> > (D4) The current implementation is built on top of HNSW graph indexing, and it is unclear how well the proposed design transfers to other graph-based ANN structures such as Vamana, NSG, or NGT. Without experiments beyond the HNSW, the generality of the method is not yet fully established.
> >
> > (D2) The paper positions itself strongly against prior projection+graph approaches, but the closest comparison line is not as directly represented in the main experiments as one would expect. In particular, it is not fully clear from the main results how much of the gain comes from the core projection-augmented idea beyond prior routing-test methods.
>
> We respond to these comments from two aspects.
>
> (1) PAG is not built on HNSW. Instead, PRT, TFB, and PES together yield a complete graph construction and search process that does not rely on any specific existing graph topology. In conclusion, PAG is a graph framework in its own right.
>
> (2) In contrast, HNSW+KS2 (Lu et al., 2025) does build KS2 on top of HNSW, although KS2 can also be applied to other graph structures such as NSSG, as shown in their paper. Therefore, to avoid the bias introduced by the underlying graph structure, we implemented KS2 (PRT-only) within the PAG framework. Our ablation study (Figs. 5, 6, 16, and 17) is designed to show the effect of each component.
>
> > (D5) Whether PAG is compatible with vector quantization such as RaBitQ, and whether the main theoretical arguments still hold in quantized vectors.
>
> (1) When PRT takes $m = 2$ and $L = d$, RaBitQ becomes a special case of PRT. However, such a parameter setting would make the index size very large without bringing any performance improvement, so we do not use it.
>
> (2) Quantization itself does not provide theoretical guarantees, so our results do not directly apply to quantized vectors. However, practitioners can still freely combine quantization with PAG for better performance.

---

> > ### Author Rebuttal · Reviewer_7gFw · 2026-04-06
> >
> > Thanks for the rebuttal. I am keeping the positive score.

---

### Official Review · Reviewer_U6d4 · 2026-03-12

**Soundness:** 3
**Presentation:** 3
**Significance:** 2
**Originality:** 2
**Overall Recommendation:** 4
**Confidence:** 4

**Summary:**

The paper introduces Projection-Augmented Graphs (PAG), a framework for high-dimensional approximate nearest neighbor search. PAG integrates random projection techniques into both graph construction and search, reducing the need for costly exact distance computations. Its effectiveness stems from three components: Probabilistic Routing Test (PRT), Test Feedback Buffer (TFB), and Probabilistic Edge Selection (PES). Experiments on six modern datasets demonstrate that PAG achieves state-of-the-art QPS-recall performance.

**Compliance With Llm Reviewing Policy:**

Affirmed.

**Ethical Review Concerns:**

Nil

**Key Questions For Authors:**

1. Can you provide experimental results or a more detailed theoretical treatment under streaming workloads that mix queries and insertions to substantiate the D6 claim?

2. Algorithm 1/2 sets maximum rounds R_max = efs/b without discussion in the main text. How was this parameter determined, and how does it impact the efficiency-recall trade-off?

3. Given a new dataset, what measurable indicators should practitioners use to predict PAG's suitability?

**Limitations:**

yes

**Strengths And Weaknesses:**

Strengths:
1. The paper is well-written, providing rigorous theoretical analysis for its core components (PRT, TFB and PES), complemented by extensive empirical evaluation and ablation studies.

2. PAG systematically integrates random projection techniques into both graph construction and search within a unified framework, with effective mechanisms (PRT, TFB and PES).

3. PAG achieves state-of-the-art QPS-recall on challenging modern datasets, demonstrating its strength on high-dimensional data and large retrieval sizes (e.g., K=1000).

Weaknesses:

1. D6 (Table 1) asserts online insertion support, but this rests solely on asymptotic complexity analysis. No experiments validate streaming workloads, and the behavior of PAG components under dynamic updates is not analyzed. It is not clear if PAG could be deployed for real-world vector databases with continuous data ingestion.

2. The working set size b = max{10, K} (Algorithm 2, line 2) and maximum rounds R_max = efs/b (Algorithm 2, line 3) appear only in pseudocode without any discussion in the main text.
The maximum rounds R_max should be a critical control parameter appearing in both construction and search algorithms and yet it lacks any explanation in the main text regarding its determination or impact on the efficiency-recall trade-off, limiting reproducibility and practical deployment.

3. PAG trails SymQG on WoltFood and GloVe (Fig. 3) and shows no advantage on legacy datasets like GIST/SIFT10M for small K (Appendix G.3). The paper attributes this post hoc to “handcrafted, sparse distributions” but offers no quantitative metrics to predict when PAG might be effective.

---

> ### Author Rebuttal · Authors · 2026-03-31
>
> We appreciate the reviewer for acknowledging our contributions and giving constructive comments.
>
> > (Weakness 1) D6 (Table 1) asserts online insertion support, but this rests solely on asymptotic complexity analysis. No experiments validate streaming workloads, and the behavior of PAG components under dynamic updates is not analyzed. It is not clear if PAG could be deployed for real-world vector databases with continuous data ingestion.
> >
> > (Question 1) Can you provide experimental results or a more detailed theoretical treatment under streaming workloads that mix queries and insertions to substantiate the D6 claim?
>
> **# Experimental Setup**
>
> To empirically evaluate the performance with online insertions, we consider the following workload.
>
> We randomly sample from the corpus 10,000 vectors as insertion queries and another 10,000 vectors as search queries. The 20,000 vectors are divided into 20 batches, each with 1,000 vectors. The insertion batches and the search batches are interleaved as a workload, with insertion as the first batch. The rest of the corpus is used to build the initial index.
>
> To make the processing times of insertion queries and search queries on the same scale, we set $efS = efC$ and tune these two parameters to control the recall.
>
> We compare PAG-Base and HNSW under $K = 100$, measuring QPS-recall for insertion and search batches on 8 datasets. Other baselines are not compared because their source codes cannot be easily modified to support this experiment.
>
> **# Experimental Results**
>
> Results can be found with the following link:
>
> - **[QPS-recall of insertion/search batches](https://anonymous.4open.science/r/PAG-A73D/assets/online-update.pdf)** (please download for higher resolution).
>
> Insertion queries are slightly slower to process than search queries for both methods. PAG-Base is much faster than HNSW in both insertion and search speeds, and PAG-Base's insertion is even faster than HNSW's search. Similar to PAG-Base's advantage in search, its speedup over HNSW in insertion can be **up to 5 times**, demonstrating its efficiency in processing online insertions.
>
> The above results will be added to the next version, along with the update to the source code.
>
> > (Weakness 2) The working set size $b = \max\{10, K\}$ (Algorithm 2, line 2) and maximum rounds $R_{\max} = efS/b$ (Algorithm 2, line 3) appear only in pseudocode without any discussion in the main text. The maximum rounds $R_{\max}$ should be a critical control parameter appearing in both construction and search algorithms and yet it lacks any explanation in the main text regarding its determination or impact on the efficiency-recall trade-off, limiting reproducibility and practical deployment.
> >
> > (Question 2) Algorithm 1/2 sets maximum rounds $R_{\max} = efS/b$ without discussion in the main text. How was this parameter determined, and how does it impact the efficiency-recall trade-off?
>
> $efS$ denotes the size of the result list for graph-based methods. A larger $efS$ leads to longer search time but higher recall. In PAG, based on the idea of TFB, we divide a large result list into multiple small buffers, each of which is processed sequentially. Here, the buffer size is set to be at least $K$, so that it is aligned with the final top-$K$ results. On the other hand, if the buffer size is too small, such as 1, TFB cannot work properly. Therefore, we set it to be at least 10 to ensure that TFB works properly. That is why $b = \max\{10, K\}$.
>
> We set $R_\max = efS/b$ because TFB is used to realize the effect of the original result list of size $efS$. Therefore, $R_{\max}$ plays exactly the same role as $efS$ in PAG.
>
> > (Weakness 3) PAG trails SymQG on WoltFood and GloVe (Fig. 3) and shows no advantage on legacy datasets like GIST/SIFT10M for small $K$ (Appendix G.3). The paper attributes this post hoc to "handcrafted, sparse distributions" but offers no quantitative metrics to predict when PAG might be effective.
> >
> > (Question 3) Given a new dataset, what measurable indicators should practitioners use to predict PAG's suitability?
>
> Whether SymQG performs well depends on whether the ranking results under the Hamming distance used by SymQG are sufficiently close to the ranking results under the exact distance. If the rankings under the two metrics are close enough, SymQG only needs to scan a very small number of points with small Hamming distances to the query in order to return the true results, which explains why SymQG performs very well in some cases. However, such approximation is often inaccurate on hard datasets (e.g., AmazonBooks, where most methods report a recall < 95%), high-dimensional settings (e.g., $d > 1024$), or when $K$ is large (e.g., $K > 100$). Such cases are common in modern datasets, compromising the practical use of SymQG.
>
> PAG is suitable for almost all high-dimensional datasets, as long as the dimension is not too small, e.g., $d \ge 100$. It works better for higher dimensions and larger retrieval sizes $K$.

---

> > ### Author Rebuttal · Reviewer_U6d4 · 2026-04-01
> >
> > Thanks for the rebuttal.  I am keeping the positive score.

---

### Official Review · Reviewer_duY7 · 2026-03-13

**Soundness:** 3
**Presentation:** 4
**Significance:** 3
**Originality:** 2
**Overall Recommendation:** 4
**Confidence:** 4

**Summary:**

The paper proposes a new graph-based algorithm for approximate nearest neighbor search. The proposed method consists of three main components: (1) a probabilistic routing test that tests whether a neighbor needs to be explored; (2) the test feedback buffer data structure that accelerates the routing test; (3) a further statistical test for probabilistic edge selection for pruning.

The authors consider six different aspects that a modern ANN algorithm should satisfy: good QPS-recall performance, fast indexing time, low memory usage, scalability to high dimensions, robustness to $k$, and support for online insertions. The authors show that their proposed method, PAG, satisfies all six aspects.

**Compliance With Llm Reviewing Policy:**

Affirmed.

**Final Justification:**

The paper presents an empirically strong method that, although could be viewed as somewhat incremental, is interesting. After the rebuttal, there still remains some concerns regarding the fairness of the experimental methodology but I remain positive about the paper as I trust the authors to perform their stated revisions.

**Key Questions For Authors:**

1. How were the hyperparameters for PAG chosen for each dataset? Was this done optimizing more for some specific metric like QPS-recall performance or indexing time for each dataset or trying to balance each of the objectives?

2. Can the authors explain in more detail why HNSW+K2 and SymphonyQG wouldn't support online updates?

**Limitations:**

Limitations are not discussed by the authors but especially limitations with respect to the experimental evaluation should be discussed. The broader impact statement is appropriate.

**Strengths And Weaknesses:**

The proposed approach builds on earlier graph + projection methods. In particular, the main performance gain of the method is due to an essentially the same test as the earlier KS2 method. The other components can also be viewed as incremental but I still think their combination is interesting and should be of practical significance especially as in this work also practical concerns like indexing time and online updates are considered using the proposed components. In the experimental results, the proposed method compares favorably to existing methods in all of the six considered aspects.

The paper is generally well executed: the results are presented appropriately, the paper is well-written, and the authors provide pseudocode and source code for their method. The experiments are performed on several modern datasets and the chosen baseline methods are appropriate.

My main concern with the article is that trying to compare the proposed method to other methods with respect to many different aspects is difficult. In particular, it is not meaningful to compare indexing time, memory consumption, and QPS all at the same time when considering only one hyperparameter setting especially for HNSW and Vamana. The hyperparameters for these methods (and SymphonyQG) explicitly control the memory usage and indexing time, e.g. using efC = 1024 is probably overly large for many of the datasets. In contrast, the hyperparameters for PAG are set differently per each dataset (Table 5).

Similarly, a lot of the results may be due to implementations and the authors don't list which implementation was used for each method. For example, the index construction in hnswlib is notoriously slow in comparison to more advanced implementations. SymphonyQG should also scale to high dimensions but the reviewer has tried the implementation and it seems more likely that there is just an implementation issue that arises on datasets with $d > 1024$ (thus all compared methods should satisfy D4). Further, measuring peak memory usage seems very implementation dependent and can lead to counterintuitive results, e.g. the memory usage of IVFPQFS and ScaNN seems to be higher than that of HNSW in the experiments but they use 4-bit quantization so with the used hyperparameters the memory usage should be lower especially on lower-dimensional datasets. I think it could be better to compare e.g. the size of the index when serialized to disk (also this is clearer if you can exclude the size of the dataset itself).

---

> ### Author Rebuttal · Authors · 2026-03-31
>
> Thank you for your insightful and supportive review.
>
> > (W1) It is not meaningful to compare indexing time, memory consumption, and QPS all at the same time when considering only one hyperparameter setting especially for HNSW and Vamana. The hyperparameters for these methods explicitly control the memory usage and indexing time, e.g. using $efC = 1024$ is probably overly large for many of the datasets. In contrast, the hyperparameters for PAG are set differently per each dataset.
>
> (1) Since the impact of $efC$ varies significantly across graph-based methods, it is difficult to determine the optimal $efC$ for each method. Nonetheless, we set $efC = 1024$ to ensure that all baselines achieve **near-optimal QPS performance** on all the datasets, because hard datasets like Word2Vec, GloVe, and AmazonBooks need a large $efC$.
>
> (2) Due to the projection-based design, PAG needs a larger $efC$ than other graph methods. For fair comparison, we set $efC \ge 1000$ for PAG-Base to avoid the indexing-time advantage that might arise  from using a smaller $efC$.
>
> (3) We show the comparison of PAG, HNSW, and Vamana under an $efC = 200$ ($L$ in Vamana) with the following external links.
>
> - **[QPS-recall](https://anonymous.4open.science/r/PAG-A73D/assets/qps-recall-small-efc.pdf)** (please download for higher resolution)
>
> - **[Indexing time](https://anonymous.4open.science/r/PAG-A73D/assets/indexing-time-small-efc.md)**
>
> The advantage of PAG in throughput and indexing time remains.
>
> > (W2.1) A lot of the results may be due to implementations and the authors don't list which implementation was used for each method.
>
> We will add links to the implementations in the revised paper.
>
> > (W2.2) SymphonyQG should also scale to high dimensions but the reviewer has tried the implementation and it seems more likely that there is just an implementation issue that arises on datasets with (thus all compared methods should satisfy D4).
>
> Although SymphonyQG (implemented by its authors) can run on datasets with $d > 1024$ (e.g., DBpedia1536, DBpedia3072, and DataCompDr), its **recalls are very low** (< 50%, see Figure 5). In this context, "support for high dimensions" should imply that the recall is practically useful; e.g., when the user wants a recall > 90%.
>
> We will revise D4 accordingly to address the confusion.
>
> > (W2.3) Measuring peak memory usage seems very implementation dependent and can lead to counterintuitive results, e.g. the memory usage of IVFPQFS and ScaNN seems to be higher than that of HNSW in the experiments but they use 4-bit quantization so with the used hyperparameters the memory usage should be lower especially on lower-dimensional datasets. I think it could be better to compare e.g. the size of the index when serialized to disk.
>
> We agree that peak memory usage is implementation dependent. Nonetheless, for the in-memory setting considered in our paper, we believe that peak memory usage is a better metric than index size to reflect the space cost. The reasons are two-fold:
>
> - For practitioners, peak memory usage is especially important because it determines whether index construction or searching can fit in the memory.
>
> -  In the search phase, due to algorithm-related factors, many graph methods such as HNSW, Vamana, and PAG incur more overhead (e.g., an array indicating visited nodes) than loading the index, making the memory usage larger than index plus dataset.
>
> For IVFPQFS and ScaNN, despite a tiny space cost of compressed vectors, they still need to access the original dataset for indexing and re-ranking, which significantly increases memory usage. The reported memory footprint thus reflects the real overhead.
>
> > (Q1) How were the hyperparameters for PAG chosen for each dataset? Was this done optimizing more for some specific metric like QPS-recall performance or indexing time for each dataset or trying to balance each of the objectives?
>
> $efC$ controls the trade-off between indexing time and throughput. $L$ and $M$ control the trade-off between space cost and throughput.
>
> In Table 5, we report two settings of PAG: PAG-Base, which targets high throughput, and PAG-Lite, which targets low space cost and low indexing time. Thus, PAG-Base uses larger values of $efC$, $M$, and $L$.
>
> Users can adjust these parameters based on their needs. We recommend $(efC, M, L) = (500, 64, \sqrt{d})$, which will be included in the revised paper.
>
> > (Q2) Can the authors explain in more detail why HNSW+K2 and SymphonyQG wouldn't support online updates?
>
> HNSW+KS2 first constructs the HNSW graph and then adds projection information based on the graph. The two steps are separate and cannot be done incrementally.
>
> SymphonyQG needs to (1) determine the navigating point, (2) start from a random graph and refine it into a similarity graph through multiple rounds, and (3) supplement edges if the out-degree does not reach the maximum value. None of the three steps is suited to online insertion.

---

> > ### Author Rebuttal · Reviewer_duY7 · 2026-04-03
> >
> > Thank you for your detailed response. I have some remaining concerns about the methodology, but I hope the authors can understand the motivation to get these details right in order to properly and fairly demonstrate the benefits of their method which looks solid.
> >
> > > (1) Since the impact of $efC$ varies significantly across graph-based methods, it is difficult to determine the optimal $efC$ for each method. Nonetheless, we set $efC = 1024$ to ensure that all baselines achieve near-optimal QPS performance on all the datasets, because hard datasets like Word2Vec, GloVe, and AmazonBooks need a large $efC$.
> >
> > The broader point I was trying to make is that it is difficult to scientifically compare recall, QPS, indexing time, and memory consumption all at the same time, especially when each method has several hyperparameters that each affect multiple of these. While I can believe that the proposed method achieves lower indexing time and memory cost, I don't think that considering a single hyperparameter combination with e.g. $efC$ set high (of course e.g. $M$ affects things as well) in order to maximize QPS and then comparing the memory usage and indexing time is enough evidence to support the broad claims made e.g. in the introduction.
> >
> > I don't in fact know what the right way would be, and I appreciate that the authors have already provided a comparison for $efC = 200$, but I'm wondering if the authors can think of additional experiments that could strengthen the support for their claims? Note that I'm not requiring that such results should be provided now during the rebuttal period.
> >
> > > Although SymphonyQG (implemented by its authors) can run on datasets with $d > 1024$ (e.g., DBpedia1536, DBpedia3072, and DataCompDr), its recalls are very low (< 50%, see Figure 5). In this context, "support for high dimensions" should imply that the recall is practically useful; e.g., when the user wants a recall > 90%.
> >
> > Right, one can try SymphonyQG e.g. on embeddings learned with Matryoshka representation learning and find out that it works fine for $d = 1024$ but the recall drops dramatically immediately when the dimensionality is set higher. To me this suggests e.g. an overflow bug in their fast scan implementation rather than an inherent flaw in the method. Anyway, I don't think it's up to the authors to find and fix a bug in another library, but I think presenting this as a flaw of the algorithm could be misleading.
> >
> > > We agree that peak memory usage is implementation dependent. Nonetheless, for the in-memory setting considered in our paper, we believe that peak memory usage is a better metric than index size to reflect the space cost.
> >
> > I don't really understand this point. The search memory usage for e.g. HNSW should be hundreds or thousands of times smaller than the memory usage of the dataset and the index. By measuring memory usage rather than serialized index size you are involving behavior related to the the allocator, page cache, shared libraries etc.
> >
> > For IVFPQFS and ScaNN, I realized that the original dataset is also kept in memory which is why I specified "especially on lower-dimensional datasets", because in that case the cost of storing the vectors with 4-bit quantization should be lower than storing the links in HNSW.
> >
> > > In Table 5, we report two settings of PAG: PAG-Base, which targets high throughput, and PAG-Lite, which targets low space cost and low indexing time.
> >
> > I was mostly curious how the hyperparameter settings for both PAG-Base and PAG-Lite were determined per dataset, because in Table 5 there are different hyperparameter settings for different datasets (for both methods).

---

> > > ### Author Response · Authors · 2026-04-05
> > >
> > > Thank you for your feedback.
> > >
> > > > (C1) I don't think that considering a single hyperparameter combination with e.g. $efC$ set high (of course e.g. $M$ affects things as well) in order to maximize QPS and then comparing the memory usage and indexing time is enough evidence to support the broad claims made e.g. in the introduction. I'm wondering if the authors can think of additional experiments that could strengthen the support for their claims?
> > >
> > > We will carefully consider your suggestion. One possible solution to this is to evaluate multiple methods under the same set of parameters such as $efC$ and $M$, as you suggested. In this sense, we will consider reporting the results by varying $efC$ (200, 400, 1000, etc.).
> > >
> > > Nonetheless, we want to emphasize that the rationale of choosing parameters for the baseline methods in the current version is seeking near-optimal QPS performance, which is often the most important metric in existing works.
> > >
> > > > (C2) To me this suggests e.g. an overflow bug in their fast scan implementation rather than an inherent flaw in the method.
> > >
> > > We will investigate this issue and possibly contact the authors of SymphonyQG for confirmation. The claim will be revised accordingly.
> > >
> > > > (C3) I don't really understand this point. The search memory usage for e.g. HNSW should be hundreds or thousands of times smaller than the memory usage of the dataset and the index. By measuring memory usage rather than serialized index size you are involving behavior related to the the allocator, page cache, shared libraries etc.
> > >
> > > We would like to give a more detailed explanation.
> > >
> > > **Indexing Phase:** First, for all the compared methods, the memory usage in the indexing phase is much larger than that in the search phase (see Fig. 4). This is not merely due to implementation issues; in fact, almost all compared methods require non-negligible additional space beyond the graph index and data storage to complete the indexing process. Therefore, peak memory consumption is the key factor that can explain why out-of-memory issues occur (e.g., SymphonyQG on MajorTOM), and we believe it is a critical metric.
> > >
> > > **Search Phase:** Although the visited array itself is not large, our point is that such additional memory overheads do exist and are often unpredictable. It is therefore difficult for users to determine where additional memory is used during query processing, and how much is required. Another example is the search phase of NSG, which requires alignment of out-degrees and introduces non-negligible additional memory overhead.
> > >
> > > To address your concern, we will revise Fig. 4 and use stacked bar charts, so that index size, dataset size, and additional overhead can be seen.
> > >
> > > > (C3) I was mostly curious how the hyperparameter settings for both PAG-Base and PAG-Lite were determined per dataset, because in Table 5 there are different hyperparameter settings for different datasets (for both methods).
> > >
> > > PAG-Base and PAG-Lite are used to illustrate two extremes: one optimized for high search performance and the other for fast indexing and low space cost. For each dataset, we choose parameter settings to represent these two extremes. The settings differ across datasets because the dimensionality and search difficulty vary significantly.
> > >
> > > Here, we take PAG-Base as an example. The parameter setting strategy is as follows:
> > >
> > > WoltFood, AmazonBooks, GloVe, and ImageNet are considered hard datasets. Word2Vec is a very hard dataset. The others are treated as normal datasets.
> > >
> > > For most normal datasets, $(efC, M) = (1000, 32)$, similar to the setting of HNSW. For most hard datasets, $(efC, M) = (2000, 64)$ for better connectivity. For Word2Vec, $efC$ is increased up to $8000$ for significant improvement on throughput.
> > >
> > > As for $L$, it depends on the dimensionality $d$ and generally increases moderately as $d$ increases. We choose from $\\{16, 32, 64, 96, 128\\}$, depending mainly on $d$.
> > >
> > > Users can adjust these parameters based on their practical requirements. The performance of PAG varies smoothly with respect to these parameters, as shown in the following figures.
> > >
> > > [QPS, $efC$ and $M$](https://anonymous.4open.science/r/PAG-A73D/assets/param-efC-M-qps.pdf)
> > >
> > > [Indexing, $efC$ and $M$](https://anonymous.4open.science/r/PAG-A73D/assets/param-efC-M-index.pdf)
> > >
> > > [QPS, $L$](https://anonymous.4open.science/r/PAG-A73D/assets/param-L-qps.pdf)
> > >
> > > [Indexing, $L$](https://anonymous.4open.science/r/PAG-A73D/assets/param-L-index.pdf)

---

### Decision · Program_Chairs · 2026-04-30

**Decision:**

Accept (regular)

**Comment:**

This submission builds upon the existing literature on graph-based nearest-neighbor search, where a heuristic is developed to filter out unnecessary pairwise distance computations in the graph construction during the indexing and in the graph search during the search (which are somewhat intertwined in graph-based schemes). The empirical evaluations highlight the proposed scheme's search time and quality, but also the improved indexing time and memory footprint.

All the reviewers acknowledge the strong performance of the proposed scheme while noting that much of the improvements are from a (potentially nontrivial) combination of mostly known techniques. There were some concerns regarding the multi-objective (QPS, recall, memory footprint, indexing time) comparison of the proposed scheme against existing baselines, where it was not clear if the advantage was due to better algorithmic improvements or due to specific implementations and inadequate hyperparameter optimization for the considered baselines. All the reviewers seemed sufficiently convinced by the author responses, and maintain a borderline accept recommendation.

My impression of the paper is similar based on my own reading of the paper and the review/rebuttal discussions. Thus I am recommending a weak accept.